# Graph Denoising Diffusion for Inverse Protein Folding

**Kai Yi** *
University of New South Wales
kai.yi@unsw.edu.au

**Bingxin Zhou** *
Shanghai Jiao Tong University
bingxin.zhou@sjtu.edu.cn

**Yiqing Shen**
Johns Hopkins University
yshen92@jhu.edu

**Pietro Liò**
University of Cambridge
pl219@cam.ac.uk

**Yu Guang Wang**
Shanghai Jiao Tong University
University of New South Wales
yuguang.wang@sjtu.edu.cn

## Abstract

Inverse protein folding is challenging due to its inherent one-to-many mapping characteristic, where numerous possible amino acid sequences can fold into a single, identical protein backbone. This task involves not only identifying viable sequences but also representing the sheer diversity of potential solutions. However, existing discriminative models, such as transformer-based auto-regressive models, struggle to encapsulate the diverse range of plausible solutions. In contrast, diffusion probabilistic models, as an emerging genre of generative approaches, offer the potential to generate a diverse set of sequence candidates for determined protein backbones. We propose a novel graph denoising diffusion model for inverse protein folding, where a given protein backbone guides the diffusion process on the corresponding amino acid residue types. The model infers the joint distribution of amino acids conditioned on the nodes' physiochemical properties and local environment. Moreover, we utilize amino acid replacement matrices for the diffusion forward process, encoding the biologically meaningful prior knowledge of amino acids from their spatial and sequential neighbors as well as themselves, which reduces the sampling space of the generative process. Our model achieves state-of-the-art performance over a set of popular baseline methods in sequence recovery and exhibits great potential in generating diverse protein sequences for a determined protein backbone structure. The code is available on https://github.com/ykiiiiii/GraDe_IF.

## 1 Introduction

Inverse protein folding, or inverse folding, aims to predict feasible amino acid (AA) sequences that can fold into a specified 3D protein structure [22]. The results from inverse folding can facilitate the design of novel proteins with desired structural and functional characteristics. These proteins can serve numerous applications, ranging from targeted drug delivery to enzyme design for both academic and industrial purposes [25, 31, 37, 52]. In this paper, we develop a diffusion model tailored for graph node denoising to obtain new AA sequences given a protein backbone.

Despite its importance, inverse folding remains challenging due to the immense sequence space to explore, coupled with the complexity of protein folding. On top of energy-based physical reasoning of a protein's folded state [1], recent advancements in deep learning yield significant progress in learning the mapping from protein structures to AA sequences directly. For example, discriminative models

---

*equal contribution.

37th Conference on Neural Information Processing Systems (NeurIPS 2023).

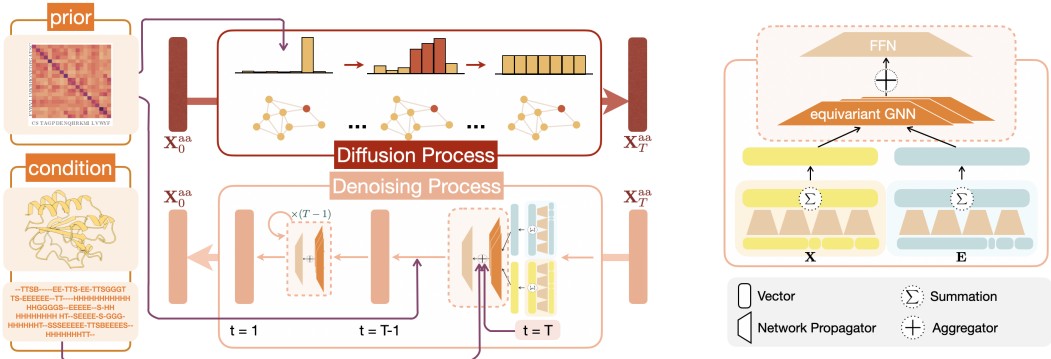

Figure 1: Overview of GRADE-IF. In the diffusion process, the original amino acid is stochastically transitioned to other amino acids, leveraging BLOSUM with varied temperatures as the transition kernel. During the denoising generation phase, initial node features are randomly sampled across the 20 amino acids with a uniform distribution. This is followed by a gradual denoising process, conditional on the graph structure and protein secondary structure at different time points. We employ a roto-translation equivariant graph neural network as the denoising network.

formulate this problem as the prediction of the most likely sequence for a given protein structure [7, 10, 17, 50]. However, they have struggled to accurately capture the one-to-many mapping from the protein structure to non-unique AA sequences.

Due to their powerful learning ability, *diffusion probabilistic models* have gained increasing attention. They are capable of generating a diverse range of molecule outputs from a fixed set of conditions given the inherent stochastic nature. For example, Torsion Diffusion [19] learns the distribution of torsion angles of heavy atoms to simulate conformations for small molecules. Concurrently, SMCDIFF [43] enhances protein folding tasks by learning the stable scaffold distribution supporting a target motif with diffusion. Similarly, DIFFDOCK [5] adopts a generative approach to protein-ligand docking, creating a range of possible ligand binding poses for a target pocket structure.

Despite the widespread use of diffusion models, their comprehensive potential within the context of protein inverse folding remains relatively unexplored. Current methods in sequence design are primarily anchored in language models, encompassing *Masked Language Models* (MLMs) [25, 23] and *autoregressive generative models* [17, 26, 28]. By tokenizing AAs, MLMs formulate the sequence generation tasks as masked token enrichment. These models usually operate by drawing an initial sequence with a certain number of tokens masked as a specific schedule and then learning to predict the masked tokens from the given context. Intriguingly, this procedure can be viewed as a discrete diffusion-absorbing model when trained by a parameterized objective. Autoregressive models, conversely, can be perceived as deterministic diffusion processes [3]. It induces conditional distribution to each token, but the overall dependency along the entire AA sequence is recast via an independently-executed diffusion process.

On the contrary, diffusion probabilistic models employ an iterative prediction methodology that generates less noisy samples and demonstrates potential in capturing the diversity inherent in real data distributions. This unique characteristic further underscores the promising role diffusion models could play in advancing the field of protein sequence design. To bridge the gap, we make the first attempt at a diffusion model for inverse folding. We model the inverse problem as a denoising problem where the randomly assigned AA types in a protein (backbone) graph is recovered to the wild type. The protein graph which contains the spatial and biochemical information of all AAs is represented by equivariant graph neural networks, and diffusion process takes places on graph nodes. In real inverse folding tasks, the proposed model achieves SOTA recovery rate, improve 4.2% and 5.4% on recovery rate for single-chain proteins and short sequences, respectively, , especially for conserved region which has a biologically significance. Moreover, the predicted structure of generated sequence is identical to the structure of native sequence.

The preservation of the desired functionalities is achieved by innovatively conditioning the model on both secondary and third structures in the form of residue graphs and corresponding node features. The major contributions of this paper are three-fold. Firstly, we propose GRADE-IF, a diffusion model backed by roto-translation equivariant graph neural network for inverse folding. It stands out

from its counterparts for its ability to produce a wide array of diverse sequence candidates. Secondly, as a departure from conventional uniform noise in discrete diffusion models, we encode the prior knowledge of the response of AAs to evolutionary pressures by the utilization of *Blocks Substitution Matrix* as the translation kernel. Moreover, to accelerate the sampling process, we adopt Denoising Diffusion Implicit Model (DDIM) from its original continuous form to suit the discrete circumstances and back it with thorough theoretical analysis.

## 2 Problem Formulation

### 2.1 Residue Graph by Protein Backbone

A residue graph, denoted as $\mathcal{G} = (\boldsymbol{X}, \boldsymbol{A}, \boldsymbol{E})$, aims to delineate the geometric configuration of a protein. Specifically, every node stands for an AA within the protein. Correspondingly, each node is assigned a collection of meticulously curated node attributes $\boldsymbol{X}$ to reflect its physiochemical and topological attributes. The local environment of a given node is defined by its spatial neighbors, as determined by the $k$-nearest neighbor ($k$NN) algorithm. Consequently, each AA node is linked to a maximum of $k$ other nodes within the graph, specifically those with the least Euclidean distance amongst all nodes within a 30Å contact region. The edge attributes, represented as $\boldsymbol{E} \in \mathbb{R}^{93}$, illustrate the relationships between connected nodes. These relationships are determined through parameters such as inter-atomic distances, local N-C positions, and a sequential position encoding scheme. We detail the attribute construction in Appendix C.

### 2.2 Inverse Folding as a Denoising Problem

The objective of inverse folding is to engineer sequences that can fold to a pre-specified desired structure. We utilize the coordinates of C$\alpha$ atoms to represent the 3D positions of AAs in Euclidean space, thereby embodying the protein backbone. Based on the naturally existing protein structures, our model is constructed to generate a protein's native sequence based on the coordinates of its backbone atoms. Formally we represent this problem as learning the conditional distribution $p(\boldsymbol{X}^{\mathrm{aa}}|\boldsymbol{X}^{\mathrm{pos}})$. Given a protein of length $n$ and a sequence of spatial coordinates $\boldsymbol{X}^{\mathrm{pos}} = \{\boldsymbol{x}_1^{\mathrm{pos}}, \ldots, \boldsymbol{x}_i^{\mathrm{pos}}, \ldots, \boldsymbol{x}_n^{\mathrm{pos}}\}$ representing each of the backbone C$\alpha$ atoms in the structure, the target is to predict $\boldsymbol{X}^{\mathrm{aa}} = \{\boldsymbol{x}_1^{\mathrm{aa}}, \ldots, \boldsymbol{x}_i^{\mathrm{aa}}, \ldots, \boldsymbol{x}_n^{\mathrm{aa}}\}$, the native sequence of AAs. This density is modeled in conjunction with the other AAs along the entire chain. Our model is trained by minimizing the negative log-likelihood of the generated AA sequence relative to the native wild-type sequence. Sequences can then be designed either by sampling or by identifying sequences that maximize the conditional probability given the desired secondary and tertiary structure.

### 2.3 Discrete Denoising Diffusion Probabilistic Models

Diffusion models belong to the class of generative models, where the training stage encompasses diffusion and denoising processes. The diffusion process $q\left(\boldsymbol{x}_1, \ldots, \boldsymbol{x}_T \mid \boldsymbol{x}_0\right) = \prod_{t=1}^{T} q\left(\boldsymbol{x}_t \mid \boldsymbol{x}_{t-1}\right)$ corrupts the original data $\boldsymbol{x}_0 \sim q\left(\boldsymbol{x}\right)$ into a series of latent variables $\{\boldsymbol{x}_1, \ldots, \boldsymbol{x}_T\}$, with each carrying progressively higher levels of noise. Inversely, the denoising process $p_\theta\left(\boldsymbol{x}_0, \boldsymbol{x}_1, ..., \boldsymbol{x}_T\right) = p\left(\boldsymbol{x}_T\right) \prod_{t=1}^{T} p_\theta\left(\boldsymbol{x}_{t-1} \mid \boldsymbol{x}_t\right)$ gradually reduces the noise within these latent variables, steering them back towards the original data distribution. The iterative denoising procedure is driven by a differentiable operator, such as a trainable neural network.

While in theory there is no strict form for $q\left(\boldsymbol{x}_t \mid \boldsymbol{x}_{t-1}\right)$ to take, several conditions are required to be fulfilled by $p_\theta$ for efficient sampling: (i) The diffusion kernel $q(\boldsymbol{x}_t|\boldsymbol{x}_0)$ requires a closed form to sample noisy data at different time steps for parallel training. (ii) The kernel should possess a tractable formulation for the posterior $q\left(\boldsymbol{x}_{t-1} \mid \boldsymbol{x}_t, \boldsymbol{x}_0\right)$. Consequently, the posterior $p_\theta(\boldsymbol{x}_{t-1}|\boldsymbol{x}_t) = \int q\left(\boldsymbol{x}_{t-1} \mid \boldsymbol{x}_t, \boldsymbol{x}_0\right) \mathrm{d}p_\theta(\boldsymbol{x}_0|\boldsymbol{x}_t)$, and $\boldsymbol{x}_0$ can be used as the target of the trainable neural network. (iii) The marginal distribution $q(\boldsymbol{x}_T)$ should be independent of $\boldsymbol{x}_0$. This independence allows us to employ $q(\boldsymbol{x}_T)$ as a prior distribution for inference.

The aforementioned criteria are crucial for the development of suitable noise-adding modules and training pipelines. To satisfy these prerequisites, we follow the setting in previous work [3]. For categorical data $\boldsymbol{x}_t \in \{1, ..., K\}$, the transition probabilities are calculated by the matrix $[\boldsymbol{Q}_t]_{ij} = q\left(\boldsymbol{x}_t = j \mid \boldsymbol{x}_{t-1} = i\right)$. Employing the transition matrix and on one-hot encoded categorical feature

$x_t$, we can define the transitional kernel in the diffusion process by:

$$q\left(x_t \mid x_{t-1}\right) = x_{t-1}Q_t \quad \text{and} \quad q\left(x_t \mid x\right) = x\bar{Q}_t, \tag{1}$$

where $\bar{Q}_t = Q_1 \ldots Q_t$. The Bayes rule yields that the posterior distribution can be calculated in closed form as $q\left(x_{t-1} \mid x_t, x\right) \propto x_t Q_t^\top \odot x\bar{Q}_{t-1}$. The generative probability can thus be determined using the transition kernel, the model output at time $t$, and the state of the process $x_t$. Through iterative sampling, we eventually produce the generated output $x_0$.

The prior distribution $p(x_T)$ should be independent of the observation $x_0$. Consequently, the construction of the transition matrix necessitates the use of a noise schedule. The most straightforward and commonly utilized method is the uniform transition, which can be parameterized as $Q_t = \alpha_t I + (1 - \alpha_t)\mathbf{1}_d \mathbf{1}_d^\top / d$ with $I^\top$ be the transpose of the identity matrix $I$, $d$ refers to the number of amino acid types (*i.e.*, $d = 20$) and $\mathbf{1}_d$ denotes the one vector of dimension $d$. As $t$ approaches infinity, $\alpha$ undergoes a progressive decay until it reaches 0. Consequently, the distribution $q(x_T)$ asymptotically approaches a uniform distribution, which is essentially independent of $x$.

# 3 Graph Denoising Diffusion for Inverse Protein Folding

In this section, we introduce a discrete graph denoising diffusion model for protein inverse folding, which utilizes a given graph $\mathcal{G} = \{X, A, E\}$ with node feature $X$ and edge feature $E$ as the condition. Specifically, the node feature depicts the AA position, AA type, and the spatial and biochemical properties $X = [X^{\text{pos}}, X^{\text{aa}}, X^{\text{prop}}]$. We define a diffusion process on the AA feature $X^{\text{aa}}$, and denoise it conditioned on the graph structure $E$ which is encoded by *equivariant neural networks* [35]. Moreover, we incorporate protein-specific prior knowledge, including an *AA substitution scoring matrix* and protein *secondary structure* during modeling. We also introduce a new acceleration algorithm for the discrete diffusion generative process based on a transition matrix.

## 3.1 Diffusion Process and Generative Denoising Process

**Diffusion Process** To capture the distribution of AA types, we independently add noise to each AA node of the protein. For any given node, the transition probabilities are defined by the matrix $Q_t$. With the predefined transition matrix, we can define the forward diffusion kernel by

$$q\left(X_t^{\text{aa}} \mid X_{t-1}^{\text{aa}}\right) = X_{t-1}^{\text{aa}} Q_t \quad \text{and} \quad q\left(X_t^{\text{aa}} \mid X^{\text{aa}}\right) = X^{\text{aa}} \bar{Q}_t,$$

where $\bar{Q}_t = Q_1 \ldots Q_t$ is the transition probability matrix up to step $t$.

**Training Denoising Networks** The second component of the diffusion model is the denoising neural network $f_\theta$, parameterized by $\theta$. This network accepts a noisy input $\mathcal{G}_t = (X_t, \mathbf{E})$, where $X_t$ is the concatenation of the noisy AA types and other AA properties including 20 one-hot encoded AA type and 15 geometry properties, such as SASA, normalized surface-aware node features, dihedral angles of backbone atoms, and 3D positions. It aims to predict the clean type of AA $X^{\text{aa}}$, which allows us to model the underlying sequence diversity in the protein structure while maintaining their inherent structural constraints. To train $f_\theta$, we optimize the cross-entropy loss $L$ between the predicted probabilities $\hat{p}(X^{\text{aa}})$ for each node's AA type.

**Parameterized Generative Process** A new AA sequence is generated through the reverse diffusion iterations on each node $x$. The generative probability distribution $p_\theta(x_{t-1}|x_t)$ is estimated from the predicted probability $\hat{p}(x^{\text{aa}}|x_t)$ by the neural networks. We marginalize over the network predictions to compute for generative distribution at each iteration:

$$p_\theta\left(x_{t-1} \mid x_t\right) \propto \sum_{\hat{x}^{\text{aa}}} q(x_{t-1}|x_t, x^{\text{aa}})\hat{p}_\theta(x^{\text{aa}}|x_t), \tag{2}$$

where the posterior

$$q\left(x_{t-1} \mid x_t, x^{\text{aa}}\right) = \text{Cat}\left(x_{t-1} \left| \frac{x_t Q_t^\top \odot x^{\text{aa}}\bar{Q}_{t-1}}{x^{\text{aa}}\bar{Q}_t x_t^\top}\right.\right) \tag{3}$$

can be calculated from the transition matrix, state of node feature at step $t$ and AA type $x^{\text{aa}}$. The $x^{\text{aa}}$ is the sample of the denoising network prediction $\hat{p}(x^{\text{aa}})$.

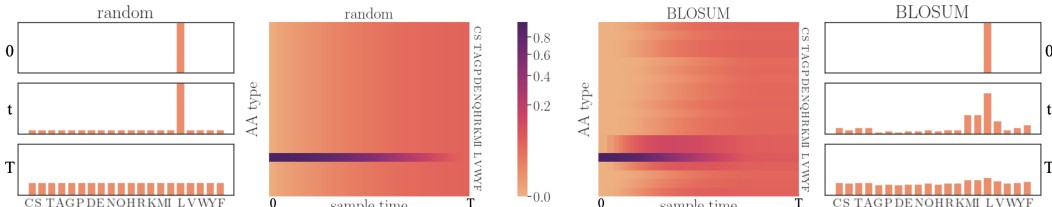

Figure 2: The middle two panels depict the transition probability of Leucine (L) from $t = 0$ to $T$. Both the uniform and BLOSUM start as Dirichlet distributions and become uniform at time $T$. As shown in the two side figures, while the uniform matrix evenly disperses L's probability to other AAs over time, BLOSUM favors AAs similar to L.

## 3.2 Prior Distribution from Protein Observations

### 3.2.1 Markov Transition Matrices

The transition matrix serves as a guide for a discrete diffusion model, facilitating transitions between the states by providing the probability of moving from the current time step to the next. As it reflects the possibility from one AA type to another, this matrix plays a critical role in both the diffusion and generative processes. During the diffusion stage, the transition matrix is iteratively applied to the observed data, which evolves over time due to inherent noise. As diffusion time increases, the probability of the original AA type gradually decays, eventually converging towards a uniform distribution across all AA types. In the generative stage, the conditional probability $p(\boldsymbol{x}_{t-1}|\boldsymbol{x}_t)$ is determined by both the model's prediction and the characteristics of the transition matrix $\boldsymbol{Q}$, as described in Equation 2.

Given the biological specificity of AA substitutions, the transition probabilities between AAs are not uniformly distributed, making it illogical to define random directions for the generative or sampling process. As an alternative, the diffusion process could reflect evolutionary pressures by utilizing substitution scoring matrices that conserve protein functionality, structure, or stability in wild-type protein families. Formally, an *AA substitution scoring matrix* quantifies the rates at which various AAs in proteins are substituted by other AAs over time [44]. In this study, we employ the Blocks Substitution Matrix (BLOSUM) [12], which identifies conserved regions within proteins that are presumed to have greater functional relevance. Grounded in empirical observations of protein evolution, BLOSUM provides an estimate of the likelihood of substitutions between different AAs. We thus incorporate BLOSUM into both the diffusion and generative processes. Initially, the matrix is normalized into probabilities using the softmax function. Then, we use the normalized matrix $\boldsymbol{B}$ with different probability temperatures to control the noise scale of the diffusion process. Consequently, the transition matrix at time $t$ is given by $\boldsymbol{Q}_t = \boldsymbol{B}^T$. By using this matrix to refine the transition probabilities, the generative space to be sampled is reduced effectively, thereby the model's predictions converge toward a meaningful subspace. See Figure 2 for a comparison of the transition matrix over time in random and BLOSUM cases.

### 3.2.2 Secondary Structure

Protein secondary structure refers to the local spatial arrangement of AA residues in a protein chain. The two most common types of protein secondary structure are alpha helices and beta sheets, which are stabilized by hydrogen bonds between backbone atoms. The secondary structure of a protein serves as a critical intermediary, bridging the gap between the AA sequence and the overall 3D conformation of the protein. In our study, we incorporate eight distinct types of secondary structures into AA nodes as conditions during the sampling process. This strategic approach effectively narrows down the exploration space of potential AA sequences. Specifically, we employ DSSP (Define Secondary Structure of Proteins) to predict the secondary structures of each AA and represent these structures using one-hot encoding. Our neural network takes the one-hot encoding as input and utilizes it to denoise the AA conditioned on it.

The imposition of motif conditions such as alpha helices and beta sheets on the search for AA sequences not only leads to a significant reduction in the sampling space of potential sequences, but also imparts biological implications for the generated protein sequence. By conditioning the sampling process of AA types on their corresponding secondary structure types, we guide the resulting protein

sequence towards acquiring not only the appropriate 3D structure with feasible thermal stability but also the capability to perform its intended function.

## 3.3 Equivariant Graph Denoising Network

Bio-molecules such as proteins and chemical compounds are structured in the 3-dimensional space, and it is vital for the model to predict the same binding complex no matter how the input proteins are positioned and oriented to encode a robust and expressive hidden representation. This property can be guaranteed by the rotation equivariance of the neural networks. A typical choice of such a network is an equivariant graph neural network [35]. We modify its SE(3)-equivariant neural layers to update representations for both nodes and edges, which reserves SO(3) rotation equivariance and E(3) translation invariance. At the $l$th layer, an Equivariant Graph Convolution (EGC) inputs a set of $n$ hidden node embeddings $\boldsymbol{H}^{(l)} = \left\{ \boldsymbol{h}_1^{(l)}, \ldots, \boldsymbol{h}_n^{(l)} \right\}$ describing AA type and geometry properties, edge embedding $\boldsymbol{m}_{ij}^{(l)}$ with respect to connected nodes $i$ and $j$, $\boldsymbol{X}^{\mathrm{pos}} = \{\boldsymbol{x}_1^{\mathrm{pos}}, \ldots, \boldsymbol{x}_n^{\mathrm{pos}}\}$ for node coordinates and $t$ for time step embedding of diffusion model. The target of a modified EGC layer is to update hidden representations $\boldsymbol{H}^{(l+1)}$ for nodes and $\boldsymbol{M}^{(l+1)}$ for edges. Concisely, $\boldsymbol{H}^{(l+1)}, \boldsymbol{M}^{(l+1)} = \mathrm{EGC}\left[ \boldsymbol{H}^{(l)}, \boldsymbol{X}^{\mathrm{pos}}, \boldsymbol{M}^{(l)}, t \right]$. To achieve this, an EGC layer defines

$$
\begin{aligned}
\boldsymbol{m}_{ij}^{(l+1)} &= \phi_e \left( \mathbf{h}_i^{(l)}, \mathbf{h}_j^{(l)}, \left\| \mathbf{x}_i^{(l)} - \mathbf{x}_j^{(l)} \right\|^2, \boldsymbol{m}_{ij}^{(l)} \right) \\
\boldsymbol{x}_i^{(l+1)} &= \mathbf{x}_i^{(l)} + \frac{1}{n} \sum_{j \neq i} \left( \mathbf{x}_i^{(l)} - \mathbf{x}_j^{(l)} \right) \phi_x \left( \mathbf{m}_{ij}^{(l+1)} \right) \\
\boldsymbol{h}_i^{(l+1)} &= \phi_h \big( \mathbf{h}_i^{(l)}, \sum_{j \neq i} \mathbf{m}_{ij}^{(l+1)} \big),
\end{aligned}
\tag{4}
$$

where $\phi_e, \phi_h$ are the edge and node propagation operations, respectively. The $\phi_x$ is an additional operation that projects the vector edge embedding $\boldsymbol{m}_{ij}$ to a scalar. The modified EGC layer preserves equivariance to rotations and translations on the set of 3D node coordinates $\boldsymbol{X}^{\mathrm{pos}}$ and performs invariance to permutations on the nodes set identical to any other GNNs.

## 3.4 DDIM Sampling Process

A significant drawback of diffusion models lies in the speed of generation process, which is typically characterized by numerous incremental steps and can be quite slow. Deterministic Denoising Implicit Models (DDIM) [39] are frequently utilized to counter this issue in continuous variable diffusion generative models. DDIM operates on a non-Markovian forward diffusion process, consistently conditioning on the input rather than the previous step. By setting the noise variance on each step to 0, the reverse generative process becomes entirely deterministic, given an initial prior sample.

Similarly, since we possess the closed form of generative probability $p_\theta(\boldsymbol{x}_{t-1}|\boldsymbol{x}_t)$ in terms of a predicted $\boldsymbol{x}^{\mathrm{aa}}$ and the posterior distribution $p(\boldsymbol{x}_{t-1}|\boldsymbol{x}_t, \boldsymbol{x}^{\mathrm{aa}})$, we can also render the generative model deterministic by controlling the sampling temperature of $p(\boldsymbol{x}^{\mathrm{aa}}|\boldsymbol{x}_t)$. Consequently, we can define the multi-step generative process by

$$
p_\theta \left( \boldsymbol{x}_{t-k} \mid \boldsymbol{x}_t \right) \propto \big( \sum_{\hat{\boldsymbol{x}}^{\mathrm{aa}}} q(\boldsymbol{x}_{t-k}|\boldsymbol{x}_t, \boldsymbol{x}^{\mathrm{aa}}) \hat{p}(\boldsymbol{x}^{\mathrm{aa}}|\boldsymbol{x}_t) \big)^T
\tag{5}
$$

where the temperature $T$ controls whether it is deterministic or stochastic, and the multi-step posterior distribution is

$$
q \left( \boldsymbol{x}_{t-k} \mid \boldsymbol{x}_t, \boldsymbol{x}^{\mathrm{aa}} \right) = \mathrm{Cat} \left( \boldsymbol{x}_{t-k} \bigg| \frac{\boldsymbol{x}_t Q_t^\top \cdots Q_{t-k}^\top \odot \boldsymbol{x}^{\mathrm{aa}} \bar{Q}_{t-k}}{\boldsymbol{x}^{\mathrm{aa}} \bar{Q}_t \boldsymbol{x}_t^\top} \right).
\tag{6}
$$

# 4 Experiments

We validate our GRADE-IF on recovering native protein sequences in **CATH** [30]. The performance is mainly compared with structure-aware SOTA models. The implementations for the main

Table 1: Recovery rate performance of **CATH** on zero-shot models.

| Model | Perplexity ↓ | | | Recovery Rate % ↑ | | | CATH version | |
|---|---|---|---|---|---|---|---|---|
| | Short | Single-chain | All | Short | Single-chain | All | 4.2 | 4.3 |
| STRUCTGNN [18] | 8.29 | 8.74 | 6.40 | 29.44 | 28.26 | 35.91 | ✓ | |
| GRAPHTRANS [18] | 8.39 | 8.83 | 6.63 | 28.14 | 28.46 | 35.82 | ✓ | |
| GCA [41] | 7.09 | 7.49 | 6.05 | 32.62 | 31.10 | 37.64 | ✓ | |
| GVP [20] | 7.23 | 7.84 | 5.36 | 30.60 | 28.95 | 39.47 | ✓ | |
| GVP-large [17] | 7.68 | 6.12 | 6.17 | 32.6 | 39.4 | 39.2 | | ✓ |
| ALPHADESIGN [9] | 7.32 | 7.63 | 6.30 | 34.16 | 32.66 | 41.31 | ✓ | |
| ESM-IF1 [17] | 8.18 | 6.33 | 6.44 | 31.3 | 38.5 | 38.3 | | ✓ |
| PROTEINMPNN [6] | 6.21 | 6.68 | 4.57 | 36.35 | 34.43 | 49.87 | ✓ | |
| PIFOLD [10] | 6.04 | 6.31 | 4.55 | 39.84 | 38.53 | 51.66 | ✓ | |
| GRADE-IF | **5.49** | **6.21** | **4.35** | **45.27** | **42.77** | **52.21** | ✓ | |

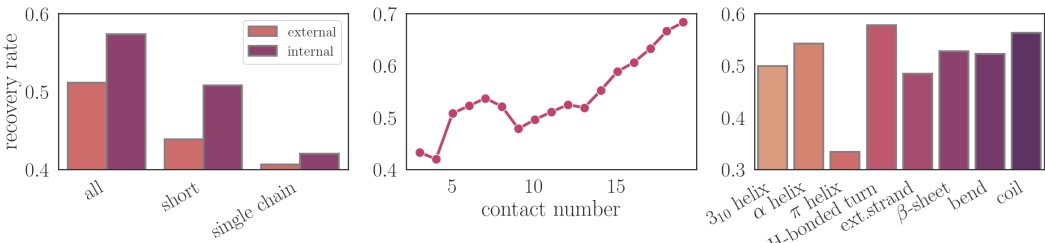

Figure 3: Recovery rate on core and surface residues and different secondary structure

algorithms (see Appendix D) at `https://github.com/ykiiiiii/GraDe_IF` are programmed with `PyTorch-Geometric` (ver 2.2.0) and `PyTorch` (ver 1.12.1) and executed on an NVIDIA® Tesla V100 GPU with $5,120$ CUDA cores and 32GB HBM2 installed on an HPC cluster.

## 4.1 Experimental Protocol

**Training Setup** We employ **CATH v4.2.0**-based partitioning as conducted by GRAPHTRANS [18] and GVP [20]. Proteins are categorized based on **CATH** topology classification, leading to a division of $18,024$ proteins for training, $608$ for validation, and $1,120$ for testing. To evaluate the generative quality of different proteins, we test our model across three distinct categories: *short*, *single-chain*, and *all* proteins. The short category includes proteins with sequence lengths shorter than 100. The single-chain category encompasses proteins composed of a single chain. In addition, the total time step of the diffusion model is configured as 500, adhering to a cosine schedule for noise [27]. For the denoising network, we implement six stacked EGNN blocks, each possessing a hidden dimension of 128. Our model undergoes training for default of 200 epochs, making use of the Adam optimizer. A batch size of $64$ and a learning rate of $0.0005$ are applied during training. Moreover, to prevent overfitting, we incorporate a dropout rate of 0.1 into our model's architecture.

**Evaluation Metric** The quality of recovered protein sequences is quantified by *perplexity* and *recovery rate*. The former measures how well the model's predicted AA probabilities match the actual AA at each position in the sequence. A lower perplexity indicates a better fit of the model to the data. The recovery rate assesses the model's ability to recover the correct AA sequence given the protein's 3D structure. It is typically computed as the proportion of AAs in the predicted sequence that matches the original sequence. A higher recovery rate indicates a better capability of the model to predict the original sequence from the structure.

## 4.2 Inverse Folding

Table 1 compares GRADE-IF's performance on recovering proteins in **CATH**, with the last two columns indicating the training dataset of each baseline method. To generate high-confidence

Table 2: Numerical comparison between generated sequence structure and the native structure.

| Method | Success | TM score | avg pLDDT | avg RMSD |
|---|---|---|---|---|
| PiFOLD | 85 | $0.80 \pm 0.22$ | $0.84 \pm 0.15$ | $1.67 \pm 0.99$ |
| PROTEINMPNN | 94 | $0.86 \pm 0.16$ | $0.89 \pm 0.10$ | $1.36 \pm 0.81$ |
| GRADE-IF | 94 | $0.86 \pm 0.17$ | $0.86 \pm 0.08$ | $1.47 \pm 0.82$ |

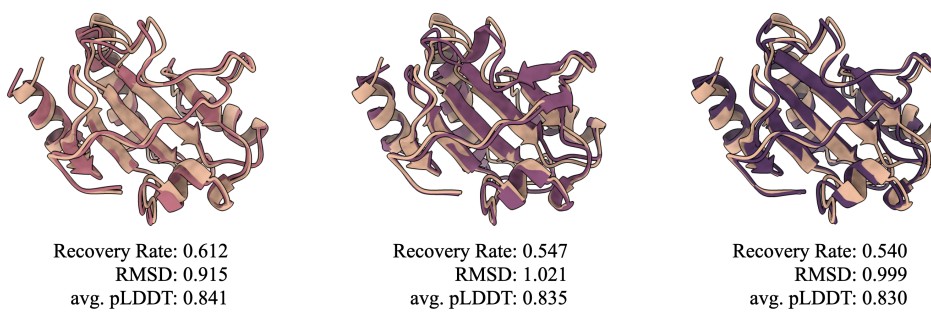

Recovery Rate: 0.612
RMSD: 0.915
avg. pLDDT: 0.841

Recovery Rate: 0.547
RMSD: 1.021
avg. pLDDT: 0.835

Recovery Rate: 0.540
RMSD: 0.999
avg. pLDDT: 0.830

Figure 4: Folding comparison of GRADE-IF-generated sequences and the native protein (in nude).

sequences, GRADE-IF integrates out uncertainties in the prior by approximating the probability $p(\boldsymbol{x}^{\mathrm{aa}}) \approx \sum_{i=1}^{N} p(\boldsymbol{x}^{\mathrm{aa}}|\boldsymbol{x}_T^i)p(\boldsymbol{x}_T^i)$. Notably, we observed an improvement of $4.2\%$ and $5.4\%$ in the recovery rate for single-chain proteins and short sequences, respectively. We also conducted evaluations on different datasets (Appendix E) and ablation conditions (Appendix F).

Upon subdividing the recovery performance based on buried and surface AAs, we find that the more conserved core residues exhibit a higher native sequence recovery rate. In contrast, the active surface AAs demonstrate a lower sequence recovery rate. Figure 3 examines AA conservation by Solvent Accessible Surface Area (SASA) (with SASA$< 0.25$ indicating internal AAs) and contact number (with the number of neighboring AAs within 8 Å in 3D space) [11]. The recovery rate of internal residues significantly exceeds that of external residues across all three protein sequence classes, with the recovery rate increasing in conjunction with the contact number. We also present the recovery rate for different secondary structures, where we achieve high recovery for the majority of secondary structures, with the exception of a minor 5-turn helix structure that occurs infrequently.

### 4.3 Foldability

We extend our investigation to the foldability of sequences generated at various sequence recovery rates. We fold generated protein sequences (by GRADE-IF) with ALPHAFOLD2 and align them with the crystal structure to compare their closeness in Figure 4 (PDB ID: 3FKF). All generated sequences are nearly identical to the native one with their RMSD $\sim 1$ Å over 139 AAs, which is lower than the resolution of the crystal structure at 2.2 Å. We have also folded the native sequence by ALPHAFOLD2, which yields an average pLDDT of $0.91$. In comparison, the average pLDDT scores of the generated sequences are $0.835$, underscoring the reliability of their folded structures. In conjunction with the evidence presented in Figure 3 which indicates our method's superior performance in generating more identical results within conserved regions, we confidently posit that GRADE-IF can generate biologically plausible novel sequences for given protein structures (See Appendix G).

The numerical investigation is reported in Table 2, where we pick the first 100 structures (ordered in alphabetical order by their PDB ID) from the test dataset and compare the performance of GRADE-IF with PROTEINMPNN and PiFOLD. We follow [48] and define the quality of a novel sequence by the TM score between its ALPHAFOLD2-folded structure and the native structure, with an above-$0.5$ score indicating the design is *successful*. Overall, our GRADE-IF exhibits high pLDDT, low RMSD, and high foldability. There are several proteins that face challenges with both GRADE-IF and baseline methods for folding with a high TM score, *i.e.*, 1BCT, 1BHA, and 1CYU, whose structural determination is based on NMR, an experimental technique that analyzes protein structure in a buffer solution. Due to the presence of multiple structures for a single protein in NMR studies, it is reasonable for folding tools to assign low foldability scores.

Table 3: Numerical comparison on diversity and recovery rate

| Method | low recovery rate | | medium recovery rate | | high recovery rate | |
|---|---|---|---|---|---|---|
| | diversity | recovery | diversity | recovery | diversity | recovery |
| PɪFᴏʟᴅ | 0.37 | 0.47 | 0.25 | 0.50 | 0.21 | 0.50 |
| PʀᴏᴛᴇɪɴMPNN | 0.51 | 0.42 | 0.27 | 0.46 | 0.26 | 0.52 |
| GʀᴀDᴇ-IF | 0.61 | 0.33 | 0.54 | 0.47 | 0.25 | 0.53 |

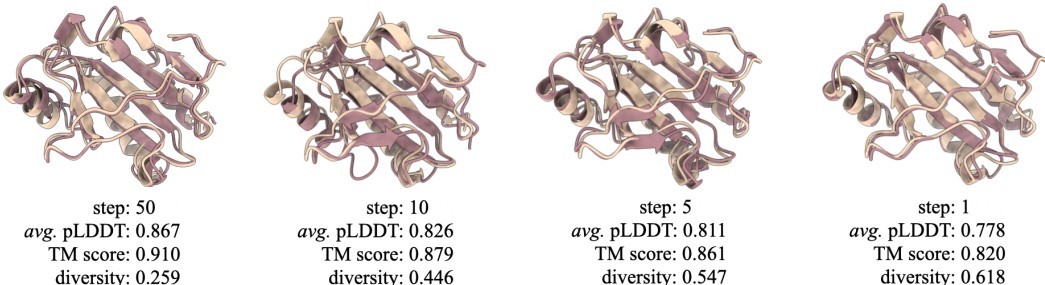

step: 50
*avg.* pLDDT: 0.867
TM score: 0.910
diversity: 0.259

step: 10
*avg.* pLDDT: 0.826
TM score: 0.879
diversity: 0.446

step: 5
*avg.* pLDDT: 0.811
TM score: 0.861
diversity: 0.547

step: 1
*avg.* pLDDT: 0.778
TM score: 0.820
diversity: 0.618

Figure 5: Folding structures of generated protein sequences with different steps.

## 4.4 Diversity

Given the one-to-many mapping relationship between a protein structure and its potential sequences, an inverse folding model must be capable of generating diverse protein sequences for a fixed backbone structure. To investigate the diversity of GʀᴀDᴇ-IF in comparison to baseline methods, we define *diversity* as $1 -$ self-similarity, where higher diversity indicates a model's proficiency in generating distinct sequences. We employ PɪFᴏʟᴅ, PʀᴏᴛᴇɪɴMPNN, and GʀᴀDᴇ-IF to generate new proteins for the test dataset at low, medium, and high recovery levels. We vary the temperature for the former two baseline methods within the range $\{0.5, 0.1, 0.0001\}$ and adjust the sample step for GʀᴀDᴇ-IF from $\{1, 10, 50\}$. The average performance from sampling 10 sequences is summarized in Table 3, revealing comparable results among the three models in both recovery rate and diversity. In general, increasing the probability temperature for PɪFᴏʟᴅ and PʀᴏᴛᴇɪɴMPNN (or decreasing the sample step in GʀᴀDᴇ-IF) leads to a reduction in uncertainty and more reliable predictions, resulting in higher diversities and lower recovery rates for all three models. While all three methods demonstrate the ability to recover sequences at a similar level, particularly with high probability temperatures (or low sample steps), GʀᴀDᴇ-IF produces significantly more diverse results when the step size is minimized. Our findings demonstrate that GʀᴀDᴇ-IF can generate sequences with a $60\%$ difference in each sample, whereas PɪFᴏʟᴅ and PʀᴏᴛᴇɪɴMPNN achieve diversity rates below $50\%$.

We next explore the foldability of these highly diverse protein sequences designed by GʀᴀDᴇ-IF. Figure 5 compares the generated proteins (folded by AʟᴘʜᴀFᴏʟᴅ2) with the crystal structure (PDB ID: 3FKF). We vary the step size over a range of $1, 5, 10, 50$, generating 10 sequences per step size to calculate the average pLDDT, TM score, and diversity. For simplicity, we visualize the first structure in the figure. Decreasing the step size results in the generation of more diverse sequences. Consequently, there is a slight reduction in both pLDDT and TM scores. However, they consistently remain at a considerably high level, with both metrics approaching $0.8$. This reduction can, in part, be attributed to the increased diversity of the sequences, as AʟᴘʜᴀFᴏʟᴅ2 heavily relies on the MSA sequences. It is expected that more dissimilar sequences would produce a more diverse MSA. Remarkably, when step$= 1$, the sequence diversity exceeds $0.6$, indicating that the generated sequences share an approximately $0.3$ sequence similarity compared to the wild-type template protein sequence. This suggests the generation of protein sequences from a substantially distinct protein family when both pLDDT and TM scores continue to exhibit a high degree of confidence.

## 5 Related Work

**Deep Learning models for protein sequence design**   Self-supervised models have emerged as a pivotal tool in the field of computational biology, providing a robust method for training extensive

protein sequences for representation learning. These models are typically divided into two categories: structure-based generative models and sequence-based generative models. The former approaches protein design by formulating the problem of fixed-backbone protein design as a conditional sequence generation problem. They predict node labels, which represent AA types, with invariant or equivariant graph neural networks [2, 17, 18, 20, 40, 42]. Alternatively, the latter sequence-based generative models draw parallels between protein sequences and natural language processing. They employ attention-based methods to infer residue-wise relationships within the protein structure. These methods typically recover protein sequences autoregressively conditioned on the last inferred AA [25, 29, 36], or employing a BERT-style generative framework with masked language modeling objectives and enable the model to predict missing or masked parts of the protein sequence [23, 26, 33, 46].

**Denoising Diffusion models** The Diffusion Generative Model, initially introduced by Sohl-Dickstein *et al.* [38] and further developed by Ho *et al.* [13], has emerged as a potent instrument for a myriad of generative tasks in continuous time spaces. Its applications span diverse domains, from image synthesis [34] to audio generation [49], and it has also found utility in the creation of high-quality animations [14], the generation of realistic 3D objects [24], and drug design [5, 43]. Discrete adaptations of the diffusion model, on the other hand, have demonstrated efficacy in a variety of contexts, including but not limited to, text generation [3], image segmentation [15], and graph generation [16, 47]. Two distinct strategies have been proposed to establish a discrete variable diffusion process. The first approach involves the transformation of categorical data into a continuous space and then applying Gaussian diffusion [4, 16]. The alternative strategy is to define the diffusion process directly on the categorical data, an approach notably utilized in developing the D3PM model for text generation [3]. D3PM has been further extended to graph generation, facilitating the joint generation of node features and graph structure [47].

## 6 Conclusion

Deep learning approaches have striven to address a multitude of critical issues in bioengineering, such as protein folding, rigid-body docking, and property prediction. However, only a few methods have successfully generated diverse sequences for fixed backbones. In this study, we offered a viable solution by developing a denoising diffusion model to generate plausible protein sequences for a predetermined backbone structure. Our method, referred to as GRADE-IF, leverages substitution matrices for both diffusion and sampling processes, thereby exploring a practical search space for defining proteins. The iterative denoising process is predicated on the protein backbone revealing both the secondary and tertiary structure. The 3D geometry is analyzed by a modified equivariant graph neural network, which applies roto-translation equivariance to protein graphs without the necessity for intensive data augmentation. Given a protein backbone, our method successfully generated a diverse set of protein sequences, demonstrating a significant recovery rate. Importantly, these newly generated sequences are generally biologically meaningful, preserving more natural designs in the protein's conserved regions and demonstrating a high likelihood of folding back into a structure highly similar to the native protein. The design of novel proteins with desired structural and functional characteristics is of paramount importance in the biotechnology and pharmaceutical industries, where such proteins can serve diverse purposes, ranging from targeted drug delivery to enzyme design for industrial applications. Additionally, understanding how varied sequences can yield identical structures propels the exploration of protein folding principles, thereby helping to decipher the rules that govern protein folding and misfolding. Furthermore, resolving the inverse folding problem allows the identification of different sequences that fold into the same structure, shedding light on the evolutionary history of proteins by enhancing our understanding of how proteins have evolved and diversified over time while preserving their functions.

## Acknowledgments and Disclosure of Funding

We thank Lirong Zheng for providing insightful discussions on molecular biology. Bingxin Zhou acknowledges support from the National Natural Science Foundation of China (62302291). Yu Guang Wang acknowledges support from the National Natural Science Foundation of China (62172370), Shanghai Key Projects (23440790200) and (2021SHZDZX0102). This project was undertaken with the assistance of computational resources and services from the National Computational Infrastructure (NCI), which is supported by the Australian Government.

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

## A  Broader Impact and Limitations

**Broader Impact**  We have developed a generative model rooted in the diffusion denoising paradigm, specifically tailored to the context of protein inverse folding. As with any other generative models, it is capable of generating *de novo* content (protein sequences) under specified conditions (e.g., protein tertiary structure). While this method holds substantial potential for facilitating scientific research and biological discoveries, its misuse could pose potential risks to human society. For instance, in theory, it possesses the capacity to generate novel viral protein sequences with enhanced functionalities. To mitigate this potential risk, one approach could be to confine the training dataset for the model to proteins derived from prokaryotes and/or eukaryotes, thereby excluding viral proteins. Although this strategy may to some extent compromise the overall performance and generalizability of the trained model, it also curtails the risk of misuse of the model by limiting the understanding and analysis of viral protein construction.

**Limitations**  The conditions imposed on the sampling process gently guide the generated protein sequences. However, in certain scenarios, stringent restrictions may be necessary to produce a functional protein. Secondary structure, as a living example, actively contributes to the protein's functionality. For instance, transmembrane $\alpha$-helices play essential roles in protein functions, such as passing ions or other molecules and transmitting a signal across the membrane. Moreover, the current zero-shot model is trained on a general protein database. For specific downstream applications, such as generating new sequences for a particular protein or protein family, it may necessitate the incorporation of auxiliary modules or the modification of training procedures to yield more fitting sequences.

## B  Non-Markovian Forward Process

We give the derivation of posterior distribution $q\left(\boldsymbol{x}_{t-1} \mid \boldsymbol{x}_t, \boldsymbol{x}^{\mathrm{aa}}\right)$ for generative process from step to step. The proof relies on the Bayes rule, Markov property, and the pre-defined transition matrix for AAs.

**Propsition 1.** *For $q\left(\boldsymbol{x}_{t-1} \mid \boldsymbol{x}_t, \boldsymbol{x}^{\mathrm{aa}}\right)$ defined in Eq 3, we have*

$$q\left(\boldsymbol{x}_{t-1} \mid \boldsymbol{x}_t, \boldsymbol{x}^{\mathrm{aa}}\right) = \mathrm{Cat}\left(\boldsymbol{x}_{t-1}\left| \frac{\boldsymbol{x}_t Q_t^\top \odot \boldsymbol{x}^{\mathrm{aa}} \bar{Q}^{t-1}}{\boldsymbol{x}^{\mathrm{aa}} \bar{Q}_t \boldsymbol{x}_t^\top}\right.\right).$$

*Proof.*  By Bayes rules, we can expand the original equation $q\left(\boldsymbol{x}_{t-1} \mid \boldsymbol{x}_t, \boldsymbol{x}^{\mathrm{aa}}\right)$ to

$$q\left(\boldsymbol{x}_{t-1} \mid \boldsymbol{x}_t, \boldsymbol{x}^{\mathrm{aa}}\right) = \frac{q\left(\boldsymbol{x}_t \mid \boldsymbol{x}_{t-1}, \boldsymbol{x}^{\mathrm{aa}}\right) q\left(\boldsymbol{x}_{t-1} \mid \boldsymbol{x}^{\mathrm{aa}}\right)}{q\left(\boldsymbol{x}_t \mid \boldsymbol{x}^{\mathrm{aa}}\right)} = \frac{q\left(\boldsymbol{x}_t \mid \boldsymbol{x}_{t-1}\right) q\left(\boldsymbol{x}_{t-1} \mid \boldsymbol{x}^{\mathrm{aa}}\right)}{q\left(\boldsymbol{x}_t \mid \boldsymbol{x}^{\mathrm{aa}}\right)}.$$

As pre-defined diffusion process, we get $q\left(\boldsymbol{x}_t \mid \boldsymbol{x}^{\mathrm{aa}}\right) = \boldsymbol{x}^{\mathrm{aa}} \bar{Q}_t$, and $q\left(\boldsymbol{x}_{t-1} \mid \boldsymbol{x}^{\mathrm{aa}}\right) = \boldsymbol{x}^{\mathrm{aa}} \bar{Q}_{t-1}$.

For the term of $q\left(\boldsymbol{x}_t \mid \boldsymbol{x}_{t-1}, \boldsymbol{x}^{\mathrm{aa}}\right)$ by Bayes rule and Markov property, we have

$$q\left(\boldsymbol{x}_t \mid \boldsymbol{x}_{t-1}, \boldsymbol{x}^{\mathrm{aa}}\right) = q\left(\boldsymbol{x}_t \mid \boldsymbol{x}_{t-1}\right) \propto q(\boldsymbol{x}_{t-1} \mid \boldsymbol{x}_t)\pi(\boldsymbol{x}_t) \propto \boldsymbol{x}_t Q_t^\top \odot \pi(\boldsymbol{x}_t)$$

where the normalizing constant is $\sum_{\boldsymbol{x}_{t-1}} \boldsymbol{x}_t Q_t^\top \odot \pi(\boldsymbol{x}_t) = (\boldsymbol{x}_t \sum_{\boldsymbol{x}_{t-1}} Q_t^\top) \odot \pi(\boldsymbol{x}_t) = \boldsymbol{x}_t \odot \pi(\boldsymbol{x}_t)$

Then $q\left(\boldsymbol{x}_t \mid \boldsymbol{x}_{t-1}, \boldsymbol{x}^{\mathrm{aa}}\right) = \frac{\boldsymbol{x}_t Q_t^\top}{\boldsymbol{x}_t}$, and the posterior distribution is:

$$q\left(\boldsymbol{x}_{t-1} \mid \boldsymbol{x}^{\mathrm{aa}}, \boldsymbol{x}_t\right) = \mathrm{Cat}\left(\boldsymbol{x}_{t-1}\left| \frac{\boldsymbol{x}_t Q_t^\top \odot \boldsymbol{x}^{\mathrm{aa}} \bar{Q}_{t-1}}{\boldsymbol{x}^{\mathrm{aa}} \bar{Q}_t \boldsymbol{x}_t^\top}\right.\right).$$

$\square$

The following gives the derivation for the discrete DDIM which accelerates the generative process.

**Propsition 2.** *For $q\left(\boldsymbol{x}_{t-k} \mid \boldsymbol{x}_t, \boldsymbol{x}^{\mathrm{aa}}\right)$ defined in Eq 6,*

$$q\left(\boldsymbol{x}_{t-k} \mid \boldsymbol{x}_t, \boldsymbol{x}^{\mathrm{aa}}\right) = \mathrm{Cat}\left(\boldsymbol{x}_{t-k}\left| \frac{\boldsymbol{x}_t Q_t^\top \cdots Q_{t-k}^\top \odot \boldsymbol{x}^{\mathrm{aa}} \bar{Q}_{t-k}}{\boldsymbol{x}^{\mathrm{aa}} \bar{Q}_t \boldsymbol{x}_t^\top}\right.\right).$$

*Proof.* By Bayes rules, we can expand the original equation $q\left(\boldsymbol{x}_{t-k} \mid \boldsymbol{x}_t, \boldsymbol{x}^{\mathrm{aa}}\right)$ to

$$q\left(\boldsymbol{x}_{t-k} \mid \boldsymbol{x}_t, \boldsymbol{x}^{\mathrm{aa}}\right) = \frac{q\left(\boldsymbol{x}_t \mid \boldsymbol{x}_{t-k}, \boldsymbol{x}^{\mathrm{aa}}\right) q\left(\boldsymbol{x}_{t-k} \mid \boldsymbol{x}^{\mathrm{aa}}\right)}{q\left(\boldsymbol{x}_t \mid \boldsymbol{x}^{\mathrm{aa}}\right)} = \frac{q\left(\boldsymbol{x}_t \mid \boldsymbol{x}_{t-k}\right) q\left(\boldsymbol{x}_{t-k} \mid \boldsymbol{x}^{\mathrm{aa}}\right)}{q\left(\boldsymbol{x}_t \mid \boldsymbol{x}^{\mathrm{aa}}\right)}.$$

As pre-defined diffusion process, we get $q\left(\boldsymbol{x}_t \mid \boldsymbol{x}^{\mathrm{aa}}\right) = \boldsymbol{x}^{\mathrm{aa}} \bar{Q}_t$, and $q\left(\boldsymbol{x}_{t-1} \mid \boldsymbol{x}^{\mathrm{aa}}\right) = \boldsymbol{x}^{\mathrm{aa}} \bar{Q}_{t-k}$.

Similarly with $q\left(\boldsymbol{x}_t \mid \boldsymbol{x}_{t-1}, \boldsymbol{x}^{\mathrm{aa}}\right)$ in Proposition 1, $q\left(\boldsymbol{x}_t \mid \boldsymbol{x}_{t-k}, \boldsymbol{x}^{\mathrm{aa}}\right) = \frac{\boldsymbol{x}_t Q_t^{\top} \cdots Q_{t-k}^{\top}}{\boldsymbol{x}_t}$ and the posterior is

$$q\left(\boldsymbol{x}_{t-k} \mid \boldsymbol{x}_t, \boldsymbol{x}^{\mathrm{aa}}\right) = \mathrm{Cat}\left(\boldsymbol{x}_{t-k} \middle| \frac{\boldsymbol{x}_t Q_t^{\top} \cdots Q_{t-k}^{\top} \odot \boldsymbol{x}^{\mathrm{aa}} \bar{Q}_{t-k}}{\boldsymbol{x}^{\mathrm{aa}} \bar{Q}_t \boldsymbol{x}_t^{\top}}\right).$$

$\square$

## C  Graph Representation of Folded Proteins

The geometry of proteins suggests higher-level structures and topological relationships, which are vital to protein functionality. For a given protein, we create a $k$-nearest neighbor ($k$NN) graph $\mathcal{G} = (\boldsymbol{X}, \boldsymbol{E})$ to describe its physiochemical and geometric properties with nodes representing AAs by $\boldsymbol{X} \in \mathbb{R}^{39}$ node attributes with 20-dim AA type encoder, 16-dim AA properties, and 3-dim AA positions. The undirected edge connections are formulated via a $k$NN-graph with cutoff. In other words, each node is connected to up to $k$ other nodes in the graph that has the smallest Euclidean distance over other nodes and the distance is smaller than a certain cutoff (*e.g.*, 30Å). Edge attributes are defined for connected node pairs. For instance, if node $i$ and $j$ are connected to each other, their relationship will be described by $\boldsymbol{E}_{ij} = \boldsymbol{E}_{ji} \in \mathbb{R}^{93}$.

The AA types are one-hot encoded to 20 binary values by $\boldsymbol{X}^{\mathrm{aa}}$. On top of it, the properties of AAs and AAs' local environment are described by $\boldsymbol{X}^{\mathrm{prop}}$, including the normalized crystallographic B-factor, solvent-accessible surface area (SASA), normalized surface-aware node features, dihedral angles of backbone atoms, and 3D positions. SASA measures the level of exposure of an AA to solvent in a protein by a scalar value, which provides an important indicator of active sites of proteins to locate whether a residue is on the surface of the protein. Both B-factor and SASA are standardized with AA-wise mean and standard deviation on the associate attribute. Surface-aware features [8] of an AA is non-linear projections to the weighted average distance of the central AA to its one-hop neighbors $i' \in \mathcal{N}_i$, *i.e.*,

$$\rho\left(\mathbf{x}_i; \lambda\right) = \frac{\left\|\sum_{i' \in \mathcal{N}_i} w_{i,i',\lambda}\left(\boldsymbol{X}^{\mathrm{pos},i} - \boldsymbol{X}^{\mathrm{pos},i'}\right)\right\|}{\sum_{i' \in \mathcal{N}_i} w_{i,i',\lambda}\left\|\boldsymbol{X}^{\mathrm{pos},i} - \boldsymbol{X}^{\mathrm{pos},i'}\right\|},$$

where the weights are defined by

$$w_{i,i',\lambda} = \frac{\exp\left(-\left\|\boldsymbol{X}_{\mathrm{pos},i} - \boldsymbol{X}_{\mathrm{pos},i'}\right\|^2 / \lambda\right)}{\sum_{i' \in \mathcal{N}_i} \exp\left(-\left\|\boldsymbol{X}_{\mathrm{pos},i} - \boldsymbol{X}_{\mathrm{pos},i'}\right\|^2 / \lambda\right)}$$

with $\lambda \in \{1, 2, 5, 10, 30\}$. The $\boldsymbol{X}^{\mathrm{pos},i} \in \mathbb{R}^3$ denotes the *3D coordinates* of the $i$th residue, which is represented by the position of $\alpha$-carbon. We also use the backbone atom positions to define the spatial conformation of each AA in the protein chain with trigonometric values of dihedral angles $\{\sin, \cos\} \circ \{\phi_i, \psi_i, \omega_i\}$.

Edge attributes $\boldsymbol{E} \in \mathbb{R}^{93}$, on the other hand, include kernel-based distances, relative spatial positions, and relative sequential distances for pairwise distance characterization. For two connected residues $i$ and $j$, the kernel-based distance between them is projected by Gaussian radial basis functions (RBF) of $\exp\left\{\frac{\|\boldsymbol{x}_j - \boldsymbol{x}_i\|^2}{2\sigma_r^2}\right\}$ with $r = 1, 2, \ldots, R$. A total number of 15 distinct distance-based features are created with $\sigma_r = \{1.5^k \mid k = 0, 1, 2, \ldots, 14\}$. Next, local frames [8] are created from the corresponding residues' heavy atoms positions to define 12 relative positions. They represent local fine-grained relations between AAs and the rigid property of how the two residues interact with each other. Finally, the residues' sequential relationship is encoded with 66 binary features by their relative position $d_{i,j} = |s_i - s_j|$, where $s_i$ and $s_j$ are the absolute positions of the two nodes in the AA chain [51]. We further define a binary contact signal [18] to indicate whether two residues contact in the space, *i.e.*, the Euclidean distance $\|C\alpha_i - C\alpha_j\| < 8$.

# D Training and Inference

In this section, we elucidate the training and inference methodologies implemented in the diffusion generative model. As shown in Algorithm 1, training commences with a random sampling of a time scale $t$ from a uniform distribution between 1 and $T$. Subsequently, we calculate the noise posterior and integrate noise as dictated by its respective distribution. We then utilize an equivariant graph neural network for denoising predictions, using both the noisy amino acid and other properties as node features, and leveraging the graph structure for geometric information. This results in the model outputting the denoised amino acid type. Ultimately, the cross-entropy loss is computed between the predicted and original amino acid types, providing a parameter for optimizing the neural network.

---
**Algorithm 1** Training

---
1: **Input**: A graph $\mathcal{G} = \{X, E\}$
2: Sample $t \sim \mathcal{U}(1, T)$
3: Compute $q(X_t | X^{\mathrm{aa}}) = X^{\mathrm{aa}} \bar{Q}_t$
4: Sample noisy $X_t \sim q(X_t | X^{\mathrm{aa}})$
5: Forward pass: $\hat{p}(X^{\mathrm{aa}}) = f_\theta(X_t, E, t, ss)$
6: Compute cross-entropy loss: $L = L_{\mathrm{CE}}(\hat{p}(X^{\mathrm{aa}}), X)$
7: Compute the gradient and optimize denoise network $f_\theta$

---

Upon completing the training, we are capable of sampling data using the neural network and the posterior distribution $p(x_{t-1} | x_t, x^{\mathrm{aa}})$. As delineated in the algorithm, we initially sample an amino acid uniformly from 20 classes, then employ our neural network to denoise $X^{\mathrm{aa}}$ from time $t$. From here, we can calculate the forward probability utilizing the model output and the posterior distribution. Through iterative processing, the ultimate model sample closely approximates the original data distribution. We can also partially generate amino acids given some known AA within the structure. Analogous to an inpainting task, at each step, we can manually adjust the prediction at known positions to the known amino acid type, subsequently introducing noise, as illustrated in the Algorithm 4.

---
**Algorithm 2** Sampling (DDPM)

---
1: Sample from uniformly prior $X_T \sim p(X_T)$
2: **for** $t$ in $\{T, T-1, ..., 1\}$ **do**
3:  Predict $\hat{p}(X^{\mathrm{aa}} | X_t)$ by neural network $\hat{p}(X^{\mathrm{aa}} | X_t) = f_\theta(X_t, E, t, ss)$
4:  Compute $p_\theta(X_{t-1} | X_t) = \sum_{\hat{X}^{\mathrm{aa}}} q(X_{t-1} | X_t, \hat{X}^{\mathrm{aa}}) \hat{p}(X^{\mathrm{aa}} | X_t)$
5:  Sample $X_{t-1} \sim p_\theta(X_{t-1} | X_t)$
6: **end for**
7: Sample $X^{\mathrm{aa}} \sim p_\theta(X^{\mathrm{aa}} | X_1)$

---

---
**Algorithm 3** Sampling (DDIM)

---
1: Sample from uniformly prior $X_T \sim p(X_T)$
2: **for** $t$ in $\{T, T-k, ..., 1\}$ **do**
3:  Predict $\hat{p}(X^{\mathrm{aa}} | X_t)$ by neural network $\hat{p}(X^{\mathrm{aa}} | X_t) = f_\theta(X_t, E, t, ss)$
4:  Compute $p_\theta(X_{t-k} | X_t) = \sum_{\hat{X}^{\mathrm{aa}}} q(X_{t-k} | X_t, \hat{X}^{\mathrm{aa}}) \hat{p}(X^{\mathrm{aa}} | X_t)$
5:  Sample $X_{t-k} \sim p_\theta(X_{t-k} | X_t)$
6: **end for**
7: Sample $X^{\mathrm{aa}} \sim p_\theta(X^{\mathrm{aa}} | X_1)$

---

**Algorithm 4** Partial Sampling
___
1: Input Mask $M$ indicate which position is fixed
2: Sample from uniformly prior $\boldsymbol{X}_T \sim p(\boldsymbol{X}_T)$
3: **for** $t$ in $\{T, T-k, ..., 1\}$ **do**
4:     Predict $\hat{p}(\boldsymbol{X}^{\text{aa}}|\boldsymbol{X}_t)$ by neural network $\hat{p}(\boldsymbol{X}^{\text{aa}}|\boldsymbol{X}_t) = f_\theta(\boldsymbol{X}_t, \boldsymbol{E}, t, ss)$
5:     Compute $\hat{p}(\boldsymbol{X}^{\text{aa}}|\boldsymbol{X}_t) = p(\boldsymbol{X}^{\text{aa}}) \odot M + \hat{p}(\boldsymbol{X}^{\text{aa}}|\boldsymbol{X}_t) \odot (1-M)$
6:     Compute $p_\theta(\boldsymbol{X}_{t-k}|\boldsymbol{X}_t) = \sum_{\hat{\boldsymbol{X}}^{\text{aa}}} q(\boldsymbol{X}_{t-k}|\boldsymbol{X}_t, \hat{\boldsymbol{X}}^{\text{aa}})\hat{p}(\boldsymbol{X}^{\text{aa}}|\boldsymbol{X}_t)$
7:     Sample $\boldsymbol{X}_{t-k} \sim p_\theta(\boldsymbol{X}_{t-k}|\boldsymbol{X}_t)$
8: **end for**
9: Sample $\boldsymbol{X}^{\text{aa}} \sim p_\theta(\boldsymbol{X}^{\text{aa}}|\boldsymbol{X}_1)$
___

## E    Inverse Folding Performance on TS50 and T500

In addition to the **CATH** dataset, we also evaluated our model using the **TS50** and **T500** datasets. These datasets were introduced by DenseCPD [32], encompassing 9888 structures for training, and two distinct test datasets comprising 50 (TS50) and 500 (T500) test datasets, respectively. The same preprocessing steps applied to the **CATH** dataset were utilized here. The denoising network comprises six sequentially arranged EGNN blocks, each boasting a hidden dimension of 256. Our model's performance, outlined in Table 4, achieved an accuracy of $61.22\%$ on **T500**, and $56.32\%$ on **TS50**, respectively.

Table 4: Recovery rate performance of **TS50** and **T500** on zero-shot models.

| Model | TS50 | | T500 | |
|---|---|---|---|---|
| | Perplexity ↓ | Recovery ↑ | Perplexity ↓ | Recovery ↑ |
| STRUCTGNN [18] | 5.40 | 43.89 | 4.98 | 45.69 |
| GRAPHTRANS [18] | 5.60 | 42.20 | 5.16 | 44.66 |
| GVP [20] | 4.71 | 44.14 | 4.20 | 49.14 |
| GCA [41] | 5.09 | 47.02 | 4.72 | 47.74 |
| ALPHADESIGN [9] | 5.25 | 48.36 | 4.93 | 49.23 |
| PROTEINMPNN [6] | 3.93 | 54.43 | 3.53 | 58.08 |
| PIFOLD [10] | 3.86 | **58.72** | 3.44 | 60.42 |
| GRADE-IF(ours) | **3.71** | 56.32 | **3.23** | **61.22** |

## F    Ablation Study

We conducted ablation studies to assess the impact of various factors on our model's performance. These elements encompassed the selection of the transition matrix (uniform versus BLOSUM), the integration of secondary structure embeddings in the denoising procedure, and the function of the equivariant neural network. As demonstrated in Figure 6, incorporating equivariance into the denoising neural network substantially enhances the model's performance. Given that the placement of protein structures in space can be arbitrary, considering symmetry in the denoising neural network helps to mitigate disturbances. Moreover, we found that including secondary structure as auxiliary information lessens uncertainty and improves recovery. Lastly, utilizing the BLOSUM matrix as the noise transition matrix boosted the recovery rate by 2%, highlighting the benefits of infusing biological information into the diffusion and generative processes. This approach reduces sample variance and substantially benefits overall model performance.

In our sampling procedure, we accelerate the original DDPM sampling algorithm, which takes every step in the reverse sampling process, by implementing the discrete DDIM as per Equation 6. This discrete DDIM allows us to skip every $k$ steps, resulting in a speed-up of the original DDPM by a factor of $k$. We conducted an ablation study on the impact of speed and recovery rate by trying different skip steps: $1, 2, 5, 10, 20, 25, 50,$ and $100$. We compare the recovery rates achieved by these different steps. Our results revealed that the recovery rate performance decays as the number of

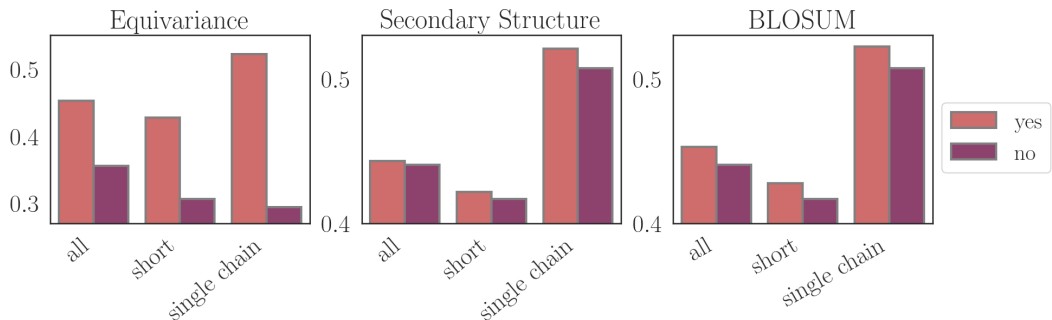

Figure 6: Recovery rate with the different selection of the transition matrix, whether considering equivariance and secondary structure.

skipped steps increases. The best performance is achieved when skipping a single step, resulting in a recovery rate of 52.21%, but at a speed of 100 times slower than when skipping 100 steps, which yields a recovery rate of 47.66%.

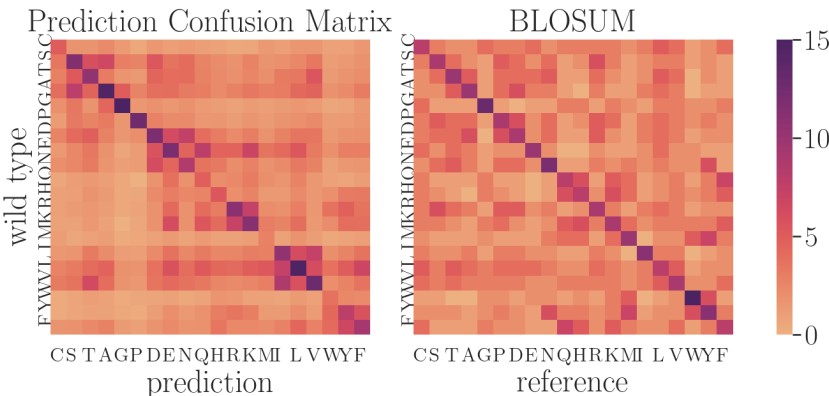

Figure 7: Comparison of the distribution of mutational prior (BLOSUM replacement matrix) and sampling results.

We further compare the diversity of GRADE-IF with PIFOLD and PROTEINMPNN in Figure 8. For a given backbone, we generate 100 sequences with a self-similarity less than 50% and employ t-SNE [45] for projection into a 2-dimensional space. At the same level of diversity, GRADE-IF encompasses the wild-type sequence, whereas the other two methods fail to include the wild-type within their sample region. Furthermore, inspiring at a recovery rate threshold of 45% for this protein, GRADE-IF manages to generate a substantial number of samples, whereas the other two methods revert to deterministic results. This further substantiates the superiority of our model in terms of achieving sequence diversity and a high recovery rate concurrently.

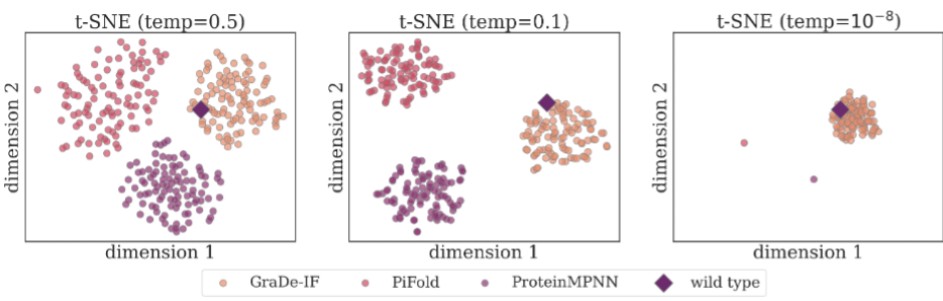

Figure 8: t-SNE of the generated sequences of GRADE-IF compared to PIFOLD and PROTEINMPNN.

# G    Additional Folding Results

We further analyzed the generated sequences by comparing different protein folding predictions. We consider the crystal structures of three native proteins with PDB IDs: **1ud9** (A chain), **2rem** (B chain), **3drn** (B chain), which we randomly choose from **CATH** dataset. For each structure, we generated three sequences from the diffusion model and used ALPHAFOLD 2 [21] to predict the respective structures. As shown in Figure 9, these predictions (in purple) were then compared with the structures of the native protein sequences (in nude). We can observe that the RMSD for all cases is lower than the preparation accuracy of the wet experiment. The results demonstrate that our model-generated sequences retain the core structure, indicating their fidelity to the original structures.

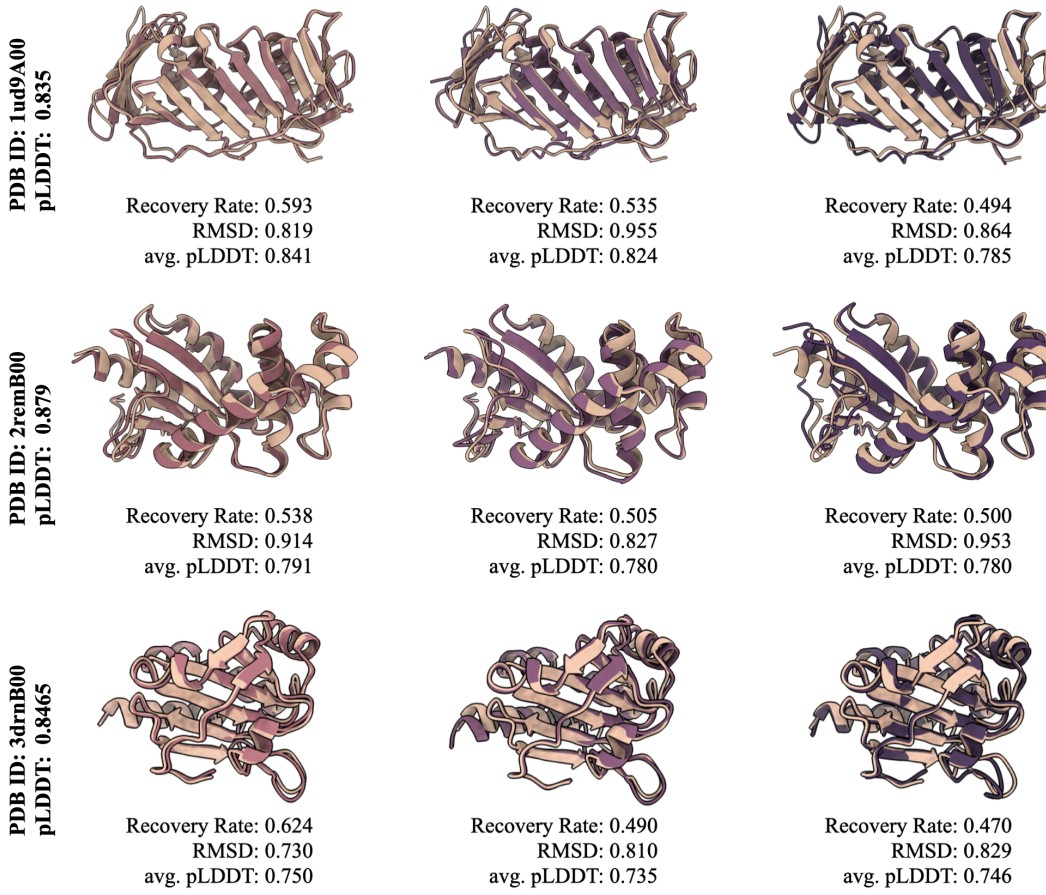

Figure 9: Folding comparisons between native sequence and generated sequence

