# Graph Denoising Diffusion for Inverse Protein Folding

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

(\boldsymbol{x}_0, \boldsymbol{x}_1, ..., \boldsymbol{x}_T) = p(\boldsymbol{x}_T) \prod_{t=1}^{T} p_\theta(\boldsymbol{x}_{t-1} \mid \boldsymbol{x}_t)$ gradually reduces the noise within these latent variables, steering them back towards the original data distribution. The iterative denoising procedure is driven by a differentiable operator, such as a trainable neural network.

While in theory there is no strict form for $q(\boldsymbol{x}_t \mid \boldsymbol{x}_{t-1})$ to take, several conditions are required to be fulfilled by $p_\theta$ for efficient sampling: (i) The diffusion kernel $q(\boldsymbol{x}_t|\boldsymbol{x}_0)$ requires a closed form to sample noisy data at different time steps for parallel training. (ii) The kernel should possess a tractable formulation for the posterior $q(\boldsymbol{x}_{t-1} \mid \boldsymbol{x}_t, \boldsymbol{x}_0)$. Consequently, the posterior $p_\theta(\boldsymbol{x}_{t-1}|\boldsymbol{x}_t) = \int q(\boldsymbol{x}_{t-1} \mid \boldsymbol{x}_t, \boldsymbol{x}_0) \, \mathrm{d}p_\theta(\boldsymbol{x}_0|\boldsymbol{x}_t)$, and $\boldsymbol{x}_0$ can be used as the target of the trainable neural network. (iii) The marginal distribution $q(\boldsymbol{x}_T)$ should be independent of $\boldsymbol{x}_0$. This independence allows us to employ $q(\boldsymbol{x}_T)$ as a prior distribution for inference.

The aforementioned criteria are crucial for the development of suitable noise-adding modules and training pipelines. To satisfy these prerequisites, we follow the setting in previous work [2]. For categorical data $\boldsymbol{x}_t \in \{1, ..., K\}$, the transition probabilities are calculated by the matrix $[\boldsymbol{Q}_t]_{ij} = q(\boldsymbol{x}_t = j \mid \boldsymbol{x}_{t-1} = i)$. Employing the transition matrix and on one-hot encoded categorical feature $\boldsymbol{x}_t$, we can define the transitional kernel in the diffusion process by:

$$q(\boldsymbol{x}_t \mid \boldsymbol{x}_{t-1}) = \boldsymbol{x}_{t-1}\boldsymbol{Q}_t \quad \text{and} \quad q(\boldsymbol{x}_t \mid \boldsymbol{x}) = \boldsymbol{x}\bar{\boldsymbol{Q}}_t, \tag{1}$$

where $\bar{\boldsymbol{Q}}_t = \boldsymbol{Q}_1 \ldots \boldsymbol{Q}_t$. The Bayes rule yields that the posterior distribution can be calculated in closed form as $q(\boldsymbol{x}_{t-1} \mid \boldsymbol{x}_t, \boldsymbol{x}) \propto \boldsymbol{x}_t\boldsymbol{Q}_t^\top \odot \boldsymbol{x}\bar{\boldsymbol{Q}}_{t-1}$. The generative probability can thus be determined using the transition kernel, the model output at time $t$, and the state of the process $\boldsymbol{x}_t$. Through iterative sampling, we eventually produce the generated output $\boldsymbol{x}_0$.

The prior distribution $p(\boldsymbol{x}_T)$ should be independent of the observation $\boldsymbol{x}_0$. Consequently, the construction of the transition matrix necessitates the use of a noise schedule. The most straightforward and commonly utilized method is the uniform transition, which can be parameterized as $\boldsymbol{Q}_t = \alpha_t \boldsymbol{I} + (1 - \alpha_t) \boldsymbol{I} \boldsymbol{I}^\top / d$ with $\boldsymbol{I}^\top$ be the transpose of the identity matrix $\boldsymbol{I}$. As $t$ approaches infinity, $\alpha$ undergoes a progressive decay until it reaches 0. Consequently, the distribution $q(\boldsymbol{x}_T)$ asymptotically approaches a uniform distribution, which is essentially independent of $\boldsymbol{x}$.

# 3 Graph Denoising Diffusion for Inverse Protein Folding

In this section, we introduce a discrete graph denoising diffusion model for protein inverse folding, which utilizes a given graph $\mathcal{G} = \{\boldsymbol{X}, \boldsymbol{E}\}$ with node feature $\boldsymbol{X}$ and edge feature $\boldsymbol{E}$ as the condition. Specifically, the node feature depicts the AA position, AA type, and the spatial and biochemical properties $\boldsymbol{X} = [\boldsymbol{X}^{\text{pos}}, \boldsymbol{X}^{\text{aa}}, \boldsymbol{X}^{\text{prop}}]$. We define a diffusion process on the AA feature $\boldsymbol{X}^{\text{aa}}$, and denoise it conditioned on the graph structure $\boldsymbol{E}$ which is encoded by *equivariant neural networks* [35]. Moreover, we incorporate protein-specific prior knowledge, including an *AA substitution scoring matrix* and protein *secondary structure* during modeling. We also introduce a new acceleration algorithm for the discrete diffusion generative process based on a transition matrix.

## 3.1 Diffusion Process and Generative Denoising Process

**Diffusion Process**  To capture the distribution of AA types, we independently add noise to each AA node of the protein. For any given node, the transition probabilities are defined by the matrix $\boldsymbol{Q}_t$. With the predefined transition matrix, we can define the forward diffusion kernel by

$$q\left(\boldsymbol{X}_t^{\text{aa}} \mid \boldsymbol{X}_{t-1}^{\text{aa}}\right) = \boldsymbol{X}_{t-1}^{\text{aa}} \boldsymbol{Q}_t \quad \text{and} \quad q\left(\boldsymbol{X}_t^{\text{aa}} \mid \boldsymbol{X}^{\text{aa}}\right) = \boldsymbol{X}^{\text{aa}} \bar{\boldsymbol{Q}}_t,$$

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

We also evaluated the speed-up sampling algorithm within this dataset, as depicted in Figure 5. As outlined in Equation equation 5, we can bypass $k$ steps during the sampling phase. We selected a range of step sizes and assessed their performance in terms of the recovery rate and the time required to sample 1200 sequences. The recovery rate mildly declines with the increment in step size, reaching 48.13% at a step size of 100. However, the sampling speed at a step size of 100 is effectively 100 times faster than at a step size of 1, demonstrating a considerable speed-up.

### 4.3 Folding Prediction on Generated Sequences

We extend our investigation to the foldability of sequences generated at various sequence recovery rates. Figure 6 contrasts the crystal structure of a native protein (PDB ID: 3FKF) with three structures folded by ALPHAFOLD2 [20], each derived from a different GRADE-IF-generated sequence. The resolution of the crystal structure stands at 2.2Å, suggesting that the folded structures of all generated sequences are nearly identical to the native one, boasting an RMSD of approximately 1-Åover 139 residues. The average pLDDT score is 0.835, which, when compared to the native protein's pLDDT of 0.91, underscores the reliability of their folded structures. In conjunction with the evidence presented in Figure 7, indicating our method's superior performance in generating more identical results within conserved regions, we confidently posit that GRADE-IF can generate biologically plausible novel sequences for given protein structures. We supplement more folding results in Appendix G.

## 5 Related Work

**Deep Learning models for protein sequence design** Self-supervised models have emerged as a pivotal tool in the field of computational biology, providing a robust method for training extensive protein sequences for representation learning. These models are typically divided into two categories: structure-based generative models and sequence-based generative models. The former approaches protein design by formulating the problem of fixed-backbone protein design as a conditional sequence generation problem. They predict node labels, which represent AA types, with invariant or equivariant graph neural networks [16, 17, 19, 40]. Alternatively, the latter sequence-based generative models

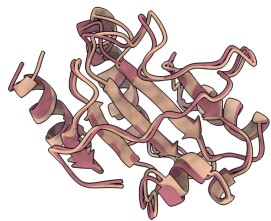 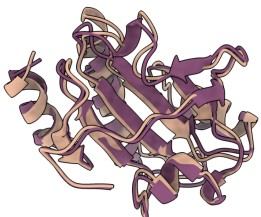 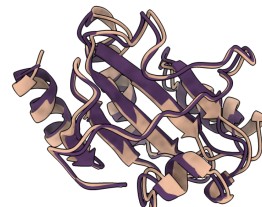

Recovery Rate: 0.612          Recovery Rate: 0.547          Recovery Rate: 0.540
RMSD: 0.915                   RMSD: 1.021                   RMSD: 0.999
avg. pLDDT: 0.841            avg. pLDDT: 0.835            avg. pLDDT: 0.830

Figure 6: Folding prediction of generated protein sequence by GRADE-IF with respect to the native protein (PDB ID: 3FKF, colored in nude).

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

5: Forward pass: $\hat{p}(\boldsymbol{X}^{\mathrm{aa}}) = f_\theta(\boldsymbol{X}_t, t, \boldsymbol{E}, ss)$
6: Compute cross-entropy loss: $L = L_{\mathrm{CE}}(\hat{p}(\boldsymbol{X}^{\mathrm{aa}}), \boldsymbol{X})$
7: Compute the gradient and optimize denoise network $f_\theta$

---

Upon completing the training, we are capable of sampling data using the neural network and the posterior distribution $p(\boldsymbol{x}_{t-1} | \boldsymbol{x}_t, \boldsymbol{x}^{\mathrm{aa}})$. As delineated in the algorithm, we initially sample an amino acid uniformly from 20 classes, then employ our neural network to denoise $\boldsymbol{X}^{\mathrm{aa}}$ from time $t$. From here, we can calculate the forward probability utilizing the model output and the posterior distribution. Through iterative processing, the ultimate model sample closely approximates the original data distribution. More importantly, we illustrate how to speed up the sampling procedure using DDIM in Algorithm 3. It can be regarded as skipping several steps in DDPM but with close performance (see Figure 5 in Section 4.2). DDPM is a special case of DDIM when skipping step $k = 1$.

---

**Algorithm 2** Sampling (DDPM)

1: Sample from uniformly prior $\boldsymbol{X}_T \sim p(\boldsymbol{X}_T)$
2: **for** $t$ in $\{T, T-1, ..., 1\}$ **do**
3:     Predict $\hat{p}(\boldsymbol{X}_0 | \boldsymbol{X}_t)$ by neural network $\hat{p}(\boldsymbol{X}_0 | \boldsymbol{X}_t) = f_\theta(\boldsymbol{X}_t, t, \boldsymbol{E}, ss)$
4:     Compute $p_\theta(\boldsymbol{X}_{t-1} | \boldsymbol{X}_t) = \sum_{\hat{\boldsymbol{X}}^{\mathrm{aa}}} q(\boldsymbol{X}_{t-1} | \boldsymbol{X}_t, \hat{\boldsymbol{X}}^{\mathrm{aa}}) \hat{p}(\boldsymbol{X}^{\mathrm{aa}} | \boldsymbol{X}_t)$
5:     Sample $\boldsymbol{X}_{t-1} \sim p_\theta(\boldsymbol{X}_{t-1} | \boldsymbol{X}_t)$
6: **end for**
7: Sample $\boldsymbol{X}^{\mathrm{aa}} \sim p_\theta(\boldsymbol{X}^{\mathrm{aa}} | \boldsymbol{X}_1)$

---

**Algorithm 3** Sampling (DDIM)

1: Sample from uniformly prior $\boldsymbol{X}_T \sim p(\boldsymbol{X}_T)$
2: **for** $t$ in $\{T, T-k, ..., 1\}$ **do**
3:     Predict $\hat{p}(\boldsymbol{X}_0 | \boldsymbol{X}_t)$ by neural network $\hat{p}(\boldsymbol{X}_0 | \boldsymbol{X}_t) = f_\theta(\boldsymbol{X}_t, t, \boldsymbol{E}, ss)$
4:     Compute $p_\theta(\boldsymbol{X}_{t-k} | \boldsymbol{X}_t) = \sum_{\hat{\boldsymbol{X}}^{\mathrm{aa}}} q(\boldsymbol{X}_{t-k} | \boldsymbol{X}_t, \hat{\boldsymbol{X}}^{\mathrm{aa}}) \hat{p}(\boldsymbol{X}^{\mathrm{aa}} | \boldsymbol{X}_t)$
5:     Sample $\boldsymbol{X}_{t-k} \sim p_\theta(\boldsymbol{X}_{t-k} | \boldsymbol{X}_t)$
6: **end for**
7: Sample $\boldsymbol{X}^{\mathrm{aa}} \sim p_\theta(\boldsymbol{X}^{\mathrm{aa}} | \boldsymbol{X}_1)$

---

## E  Inverse Folding Performance on TS50 and T500

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

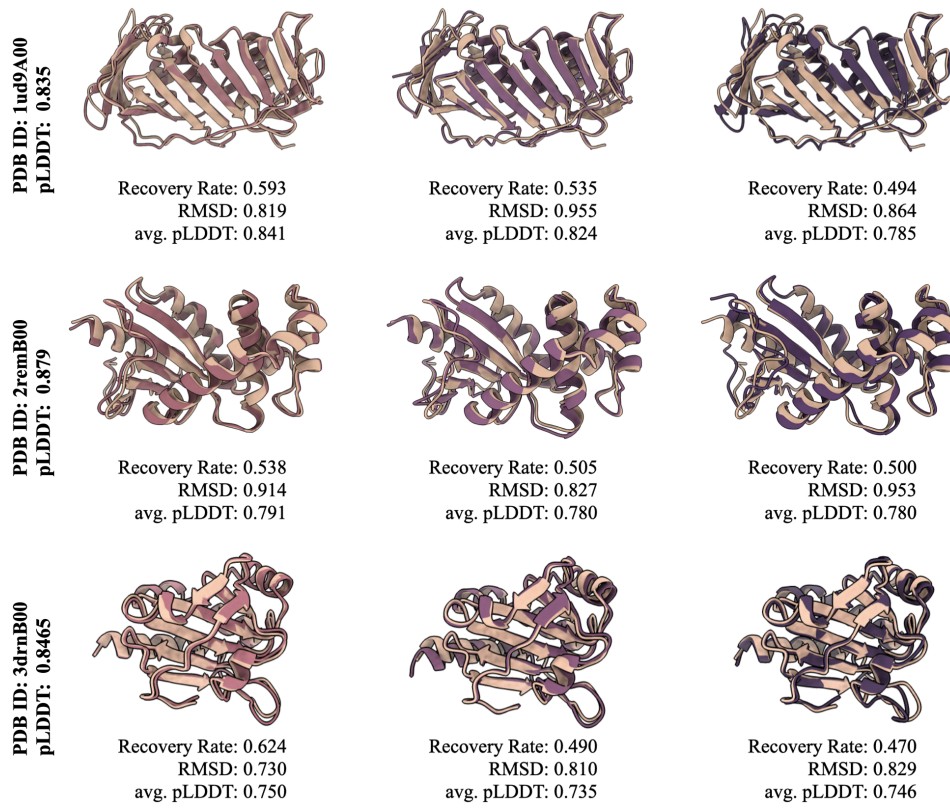

Figure 9: Folding comparsion between native sequence and generated sequence