# OpenReview forum: "Graph Denoising Diffusion for Inverse Protein Folding"
_NeurIPS.cc/2023/Conference — NeurIPS 2023 poster_

### Official Review · Reviewer_92pD · 2023-06-25

**Soundness:** 2 fair
**Presentation:** 4 excellent
**Contribution:** 3 good
**Rating:** 6
**Confidence:** 5

**Summary:**

The paper proposes discrete diffusion for the task of inverse protein folding (IPF). The many-to-one mapping of sequence to structure warrants a generative model of sampling multiple possible sequences that fold into a given structure. Nearly all prior works use autoregressive generative models so the exploration of non-autoregressive models is high interest for its improvements in accuracy and speed. The authors propose several IPF specific improvements to discrete diffusion that results in state-of-the-art performance on commonly used benchmarks. Additional analysis is performed on accuracy for different classes of residues such as surface exposed, diversity, and designability.

The contributions can be summarized as follows:

- First exploration of discrete diffusion for IPF.
- Discrete diffusion improvements such as a BLOSUM weighted transition kernel, secondary structure auxiliary prediction, DDIM for faster sampling.
- New SOTA performance on IPF benchmarks.

**Strengths:**

I enjoy the direction and development of ideas in this paper. Protein sequences are largely driven by their local environment based on the graph construction. It makes a lot sense to free the decoding order to be learned rather than left-to-right. Though the method is not new, the application is novel and well motivated. I appreciate the authors exploring this direction for IPF.

- Well-motivated by taking advantage of the graph structure inherent in the problem to improve sampling.
- Well written. I could follow the math and figures to understand the method. I appreciate the authors shared code.
- Removes limitations of left-to-right decoding order of previous works which results in higher diversity.

**Weaknesses:**

- Comparison to alternative discrete diffusion methods. I hoped for more discussion on the merits and considerations of choosing D3PM over alternative methods of discrete diffusion such as analog bits and latent diffusion.

- No in-depth analysis of foldability. There is a question of whether higher recovery rate is always better. For instance, foldingdiff [1] showed in Table S1 and S2 that foldability was much higher with ProteinMPNN than with ESM-1F -- the latter which has higher sequence recovery. Including a benchmark on foldability and not just a few examples would be important to understand foldability as a function of sequence recovery on a benchmark would be high importance to understand the relationship. I already believe the contributions are strong and this is merely a suggestion for higher impact.

- No explanation of different benchmark numbers. I noticed the numbers for proteinmpnn don't match what is reported in this paper.


[1] https://arxiv.org/abs/2209.15611
[2] https://www.science.org/doi/10.1126/science.add2187
[3] https://www.biorxiv.org/content/10.1101/2022.04.10.487779v1


**Questions:**

- As mentioned above, what is the relationship between sequence recovery and designability/foldability? [note to self of checking their foldability results.

- Why does the method only use C-alpha positions? ProteinMPNN found using all backbone atoms to improve results. I would be curious if all backbone atoms helps and if noise augmentation improves.

- Have the authors tried not providing hand-crafted geometric features? I would’ve suspected these can be inferred from the raw coordinates like in ProteinMPNN.

- Why do the benchmark numbers not match those reported in ProteinMPNN?

- There is are some typos and weird language I noticed.
    - Line 78. “Meticulously curated” is a weird phrase. I would reword it.
    - Line 88. Needs capitalization.
    - Line 93. x_1^{pos} is bold face while the rest are not.
    - Line 134. Equivariant is misspelled.
    - Line 143. normalized is misspelled.

**Limitations:**

- There no limitations discussed. It would be interesting to see where the method fails and initial preliminary thoughts on how it can be improved.

---

> ### Author Rebuttal · Authors · 2023-08-09
>
> Thank you very much for recognizing the novelty and contribution of our work. Your insightful comments helped us enrich the analysis a lot. By the response below we hope we address your concerns properly.
>
> **weaknesses**:
>
> 1. *comparison to alternative diffusion methods*: The question of choosing between discrete diffusion in original space (e.g., D3PM) and latent continuous space diffusion (e.g., analog bits) remains open. Each approach has its merits and limitations. Discrete diffusions provide transparency in the generative process, enabling observation of sequence transformation, and latent space diffusion benefits from established continuous diffusion methods used in image/video generation. The latter allows a similar treatment of discrete data diffusion within a continuous Gaussian diffusion framework.
>
> 2. *foldability*: Thank you for highlighting this important aspect. We've incorporated your suggestion by adopting the TM score to assess foldability. We've provided Table 4 in the PDF, summarizing average TM scores (with standard deviations) between structures predicted by AlphaFold2 for generated sequences and their wild-type counterparts. GraDe-IF consistently achieves the highest TM scores with the least variance. Moreover, we've extended foldability evaluation to a wider set of 42 proteins (Table 5 in PDF), confirming GraDe-IF's superiority in terms of foldability.
>
> 3. *different benchmark performance*: The benchmark discrepancies arise due to our distinct dataset split rules. Although both ProteinMPNN and our study utilize CATH4.2, ProteinMPNN used random splitting by 80:%10%:10% (according to their experimental details), and we follow Ingraham et al.'s [1] split, ensuring no overlap in CATH topology classification between training, validation, and test sets. This explains the divergent reported results. Same as [1,2,3], we implemented the latter splitting rule for our method. We have also tested the 80:%10%:10% splitting rule as established in ProteinMPNN’s paper, and it returns a higher recovery rate on the test set at 59.6%. As a reference, ProteinMPNN reported their recovery rate at 50.8%. Having said that, we would like to emphasize again that we do not optimize our diffusion model towards a close-to-1 recovery rate of the generated sequence. Instead, we expect a preferred model to perform higher diversity on the generated sequences while capturing the composition of amino acids at those conserved regions or positions (such as the buried regions in proteins.).
>
> **Questions**:
>
> 1. *foldability*:  see weaknesses point 2.
>
> 2. *coordinates*: (1) The objective of the designed model is to generate possible amino acid sequences for a given protein backbone. To this end, we make all the operations, including the diffusion process and the message passing propagators on residue levels. Since the atom types for different amino acids only differ on the side chain, atom types on the backbone are determined. Regarding the atom positions, we initially utilized C-alpha positions to represent the position of the residue, which is a widely applied choice for generating residue-level protein graphs. (2) While we did not employ the position of other atoms explicitly, we did include their spatial relationship by defining the relative position and local frames of local atoms. The detailed definitions were introduced in Appendix C.
>
> 3. *hand-crafted features*: We agree that it is possible to infer the processed node features from the raw coordinates. However, there is no doubt that a much higher volume of input proteins would be required to train a more sophisticated (and potentially more powerful) model. Moreover, defining hand-crafted features has been used in several previous protein-related works, which indicates that it can be considered a promising choice for modeling protein topology. While we remain open to exploring this idea in future research, we conducted a preliminary ablation study that investigates the influence of the dihedral angle on the model's performance. The study revealed that omitting this hand-crafted feature led to a decrease in the recovery rate of the entire test dataset, from 52.21% to 51.47%, and an increase in perplexity from 4.35 to 4.58.
>
> 4. *mismatched performance for ProteinMPNN*: see weaknesses point 3.
>
> 5. *typos*: We've diligently corrected the typos you pointed out in the revised version.
>
> We have followed your insightful comments and suggestions to address the concerns you raised and polish our work. We believe these changes have greatly enhanced the quality and contribution of this paper. We believe that the additional analysis and discussion now better support the merit of our work. We would be more than willing to take your further suggestions, and we kindly request you reconsider the score you assigned to our paper.
>
>
> **Reference**:
>
> [1] Ingraham, John, et al. "Generative models for graph-based protein design." In Advances in neural information processing systems (2019).
>
> [2] Gao, Zhangyang, Cheng Tan, and Stan Z. Li. "PiFold: Toward effective and efficient protein inverse folding." In International Conference on Learning Representations (2022).
>
> [3] Zheng, Zaixiang, et al. "Structure-informed Language Models Are Protein Designers." In International Conference of Machine Learning (2023).

---

> > ### Comment · Reviewer_92pD · 2023-08-10
> > **Response**
> >
> > Thank you the response. I believe the method is technically sound. My score is from my concerns in the evaluation.
> >
> > > ProteinMPNN used random splitting by 80:%10%:10% (according to their experimental details),
> >
> > I am not sure this is true. Their experimental details state "we used a set of 19.7k high resolution single-chain structures from the PDB were split into train, validation and test sets (80/10/10) based on the CATH4.2 40% non-redundant set of proteins (1, 8)." where reference (1) is Ingraham et al. It seems to me ProteinMPNN and Ingraham follow the same splits. Can the authors clarify how they arrived at their conclusion?
> >
> > > we've extended foldability evaluation to a wider set of 42 protein
> >
> > I appreciate the addition of more proteins. I commented in the global comment asking how these 42 were selected. Are they in the test set?

---

> > > ### Author Response · Authors · 2023-08-13
> > > **Re-Reviewer 92pD: reults on ProteinMPNN and 42 proteins**
> > >
> > > We appreciate your meticulous attention to detail and the insightful comparisons you've made regarding the dataset split methodology. Regarding your concerns:
> > >
> > > 1. **Performance of ProteinMPNN**:
> > > (1) Indeed, Ingraham et al. [1] take the random split of the training, validation, and test sets by an 80/10/10 ratio, along with the careful consideration of protein CATH categories to remove redundancy, which has been a common practice in subsequent studies as well. However, we did not find explicit rules for dataset processing in ProteinMPNN's paper, specifically concerning the removal of overlap data. (2) Our further investigation into the ProteinMPNN's GitHub repository, including its discussion board, revealed an interesting detail. ProteinMPNN employed a random decoding technique to sample sequences, leading to an improvement in the recovery rate. In our initial implementation, we adhered to ProteinMPNN's default settings, which utilized the same input sequence order for decoding and yielded suboptimal results in terms of CATH recovery rate. By incorporating the random decoding strategy, we were able to enhance ProteinMPNN's result with a higher recovery rate of 49.9% and a perplexity of 4.576. We acknowledge the discrepancy in the methodologies employed and extend our gratitude for your keen attention to this crucial aspect. We will make clarifications and update this result in our revision.
> > >
> > > 2. **42 Proteins**:
> > > All the proteins are from the test dataset. As we have replied in the global comment, we fold the first 42 proteins in the test dataset in alphabetical order by their PDB ID. Due to time limitations, we were only able to fold this subset of proteins for the generated protein sequences by the three models compared. We are currently in the process of conducting evaluations on the remaining proteins in the test set. We are committed to providing updated and comprehensive scores for a broader range of proteins.

---

> > > > ### Comment · Reviewer_92pD · 2023-08-14
> > > > **Response**
> > > >
> > > > Thank you for both the investigation and clarification. I appreciate the investigation and correction of ProteinMPNN's performance. I would have preferred to see foldability performance on a larger set of proteins but understand the time constraint. I am between a 5 and 6 due to the limited foldability but will put it as a 6. I would have put it higher if the initial foldability assessment was strong. To the AC, I am recommending weak acceptance on the condition a larger set of proteins is included in the final evaluation.

---

### Official Review · Reviewer_ZARz · 2023-07-02

**Soundness:** 3 good
**Presentation:** 3 good
**Contribution:** 3 good
**Rating:** 5
**Confidence:** 4

**Summary:**

The paper highlights that "Existing discriminative models struggle to capture the diverse range of solutions, while generative diffusion probabilistic models offer the potential for generating diverse sequence candidates." They propose to use the denoise diffusion model together with the prior information from the secondary structure and BLOSUM matrix to train an inverse folding model. As shown in the experiments, their method achieves better results than the previous baselines, and they also show the folding results for the model predictions.

**Strengths:**

[+] The proposed graph denoising diffusion model introduces a new perspective for addressing the challenging problem of inverse protein folding. It leverages the power of diffusion probabilistic models to generate a diverse set of sequence candidates for a given protein backbone. In protein design, diversity is a key point, since we want diverse candidates to build a library for wet-lab experiments.
[+] The model achieves state-of-the-art performance compared to popular baseline methods in sequence recovery.
[+] The method includes prior information during the model training, and the method is easy-to-follow.
[+] The authors also check SASA and other properties, which is useful for protein design and engineering.

**Weaknesses:**

[-] Although the paper mentions diversity, however, they mainly measure the recovery ratio (wildtype accuracy) and the PPL, which cannot tell the readers whether their model captures the "useful" diversity. The diversity is only shown in Figure 4, and I think it is not enough to demonstrate the model's advances in diversity.
[-] The baseline PiFold or ESMFold algorithm can also introduce diversity by controlling the temperature and do sampling based on the probability score, a detailed comparison with the baselines could improve the experiments.


**Questions:**

1. The authors show the trade-off between speed and recovery ratio. Here, I wonder if the authors use a standard Euler solver or any other sampling schedule to do the inference.
2. In Figure 6, the authors show the results of several PDB proteins,  do the authors make sure that these proteins have < 30% sequence similarity as the proteins in the training dataset?

---

> ### Author Rebuttal · Authors · 2023-08-08
>
> We appreciate your thoughtful feedback on our paper and recognizing its contribution and performance. We have taken your remarks into careful consideration and offer the following responses to your concerns and questions:
>
> **weaknesses**:
> - *measurement of diversity, and additional comparison with PiFold and ProteinMPNN*: (1) The table below compares the recovery rate and diversity of the generation results by controlling temperature = {0.5, 0.1, 0.0001} for PiFold and ProteinMPNN and sample step ={20, 100, 250} for GraDe-IF. The three levels are divided roughly into low, medium, and high recovery rates in the table below. The results, quantified as average diversity and recovery rate, are depicted in the table provided. For clarity, the metric "diversity" is quantitatively defined as `diversity = 1-sequence identity`, where the average sequence identity is computed pairwise for generated sequences. For instance, in the first cell of the table, where the method is PiFold, the recovery level is low, and the metric is diversity, the diversity value is indicated as 0.3796.
> (2) In response to the task of aligning recovery rate levels, we have introduced an intermediary plot positioned between the two subfigures in Figure 4 (refer to Figure 1 in the attached PDF document). This supplementary plot effectively illustrates the intricate relationship between recovery rates and diversity across the three methods on a finer scale. Notably, the visualization highlights the rapid contraction of the sampling space for both PiFold and ProteinMPNN. In contrast, the samples generated by GraDe-IF exhibit a broader distribution that encompasses the wild type.
>
> |    | low recovery | | medium recovery | | high recovery  |  |
> |------------------|:-------:|:--------:|:-----:|:-------:|:------:|:------:|
> |   (Method)  | (diversity) | (recovery) |  (diversity) | (recovery) | (diversity) | (recovery) |
> | PiFold	| 0.3796 | 0.4794 | 0.2535 | 0.5084 | 0.2181 | 0.5087 |
> | ProteinMPNN | 0.5159 | 0.4268 | 0.2735 | 0.4679 | 0.2657 | 0.4679 |
> | GraDe-IF | 0.5899 | 0.4142 | 0.5462 | 0.4586 | 0.5296 | 0.4755 |
>
>
> **Questions**:
> - *sampling schedule for inference*: While the standard Euler solver was not implemented in our study due to the distinct characteristics of our diffusion process, which involves a discrete transition matrix that does not follow a continuous stochastic differential equation, we acknowledge the potential for future exploration. In forthcoming research, there may be merit in formulating a score-based model and developing a sampling algorithm that operates within a continuous time framework. It's plausible that continuous solvers, such as the Euler solver, could play a role in this endeavor.
> - *sequence similarity of the generated protein sequences*: We have meticulously examined the generated sequences, confirming that none of them exhibit a sequence identity exceeding 30% with the training samples. It is essential to emphasize that the intention behind presenting Figure 6 is to showcase the capacity of our generated protein sequences, conditioned on a template protein backbone, to reliably fold back to the template structure (RMSD < resolution) with a high degree of confidence (pLDDT > 0.8). We want to underline that all the displayed proteins, including 3FKF in Section 4, along with 1ud9, 2rem, and 3drn in Appendix G, were selected from the test set, which was properly processed to ensure no overlapping with the training or validation set. Moreover, every visualized sequence underwent a thorough comparison against the wild-type template protein sequence, yielding sequence identity values that fall within the 40% to 60% range.

---

### Official Review · Reviewer_j843 · 2023-07-04

**Soundness:** 2 fair
**Presentation:** 2 fair
**Contribution:** 3 good
**Rating:** 6
**Confidence:** 3

**Summary:**

The authors of the manuscript present GraDE-IF, a diffusion model based method for inverse protein folding given the backbone of the structure. The denoising network is a graph neural network that's equivariant to rotations and translations. Moreover, a biologically relevant inductive bias is incorporated into the discrete diffusion process by replacing the uniform amino acid transition probabilities with amino acid substitution scoring matrices. Finally, sampling is accelerated by deploying a variant of the Denoising Diffusion Implicit Model (DDIM).

**Strengths:**

To the best of my knowledge, this is the first work that explores the use of a discrete diffusion model for inverse protein folding. The fact that prior biological knowledge is incorporated directly into the discrete diffusion process increases the appeal of this method. The presented results look promising.

**Weaknesses:**

Even though I enjoyed reading the paper, there are still some concerns.

First and foremost, in my opinion the reported metrics do not fully support the claim that the model is capable of generating diverse protein sequences. Figure 6 shows some qualitative examples of different sequences leading to plausible structures, but I'm missing quantitative results on diversity/uniqueness, and an average RMSD/pLDDT for a large batch of generated sequences folded by AlphaFold2 for different proteins. This would make the story much more convincing.

Additionally, the paper is quite sloppy in some places, with confusing mathematical notation, wrong figure references, and occasionally poor sentence constructions. I appreciate that this is likely due to time constraints, but some time should be spent on fixing this for the final paper in case it gets accepted.

I will address the more minor concerns in the "Questions" part.

**Questions:**

1. As mentioned in "Weaknesses", I missed some kind of quantitative metric to measure diversity / uniqueness of generated sequences accross proteins.
2. Another valuable addition to the results, although it might not be possible due to time constraints, would be to show how this method could aid protein design, e.g. by providing a synthetic backbone and generating plausible corresponding sequences.
3. Figure 1: the text in the "prior" and "condition" blocks is quite small. For me this caused some confusion because at a quick glance it seemed like the bottom part of "condition" was an amino acid sequence rather than secondary structure annotation.
4. Table 1: Are the CATH versions correct? The first sentence of section 4.1 (line 233) mentions CATH v4.3.0.
5. Figure 4: The 45% threshold seems quite conveniently chosen such that only one sequence remains for the other two models. It would be informative to include this plot for some different recovery rate cut-offs, if not in the main text then at leats in the supplementary. Additionally, for the figure on the right, no unit is given for "speed", and there is no comparison to other models (PiFold, ProteinMPNN).
6. Missing references/proofs to support the connections made in line 45-51.
7. Missing defenitions:
    * FFN (Figure 1): mention feed forward network somewhere.
    * $\mathcal{E}$ (line 76) is never defined.
    * $d$ (line 126) is never defined (number of categories).
    * $\mathbf{A}$ (line 131) is never defined.
8. Confusing mathematical notation:
    * The graph notation seems inconsistent accross the paper.
    * I think there's a "pos" superscript missing in line 93 (i.e. $\mathbf{X} \rightarrow \mathbf{X^{pos}}$)
    * The expression in line 126 seems a bit confusing. Isn't the transpose of the identity matrix the same (i.e. $\mathbf{I}^T=\mathbf{I}$) and wouldn't $\mathbf{I} \ \mathbf{I}^T=\mathbf{I}$? It might be that this is the point you wanted to make, but it was not completely clear to me.
9. Wrong figure references line 260, line 289.
10. Not all appendices are referenced to in the main text.
11. A grammar check would improve the flow of the paper.

**Limitations:**

The authors adequately dicuss the limitations of their work in the appendix, even though I would prefer it if some of this discussion was transferred to the main paper.

---

> ### Author Rebuttal · Authors · 2023-08-09
>
> Thank you for your meticulous feedback, as well as your recognition of the novelty of our work. Below we respond to your concerns and questions point-by-point.
>
> **Weaknesses** :
>
> 1. *Quantitative Results on Diversity*: We have incorporated quantitative evaluations of generated results using the `diversity` metric, defined as `1 - sequence identity`. We extended our assessments to average RMSD and pLDDT across a broader spectrum of 42 distinct proteins. Note that the reason we not investigating a larger volume of proteins is because that AlphaFold failed to fold more structures within the limited time for rebuttal. (1) Table 1 in PDF compares the recovery rate and diversity of the generation results by controlling temperature = {0.5, 0.1, 0.0001} for PiFold and ProteinMPNN and sample step ={20, 100, 250} for GraDe-IF. The three levels are divided roughly into low, medium, and high recovery rates in the table below. The results, quantified as average diversity and recovery rate, are depicted in Table 1 in PDF.  (2) We expanded the validation of RMSD and pLDDT across a wider selection of 42 proteins and reported the results in Table 2 in PDF. Notably, GraDe-IF exhibited the highest average pLDDT and lowest RMSD, highlighting its effectiveness over baseline methods.
>
> 2. *Typo*: We have carefully gone through the main text to fix the typos and inappropriately defined notations, combining suggestions from all the reviewers.
>
> **Questions**:
>
> 1. *quantitative metrics on the generated results*: Refer to the response provided above.
> 2. *sequence generation for synthetic backbone*: Thank you for suggesting a new possibility for our design. There is no doubt that generating plausible sequences for synthetic backbones is of great significance for designing structural proteins. While the limited time for rebuttal prevents us from preparing a throughout investigation upon generating valid sequences for practically meaningful protein structures, it opens a promising direction for future investigation, combining in silico validation and wet experiments through structural characterization methods like NMR, X-ray crystallography, and cryo-EM.
> 3. *conditions in Figure 1*: We have fixed the problem in the revision by magnifying the contents in both prior and condition blocks, mitigating any residual confusion.
> 4. *CATH versions*: We utilized CATH4.2 for both training and evaluation. Thank you for highlighting this inconsistency. We have fixed the typo in the revised version.
> 5. *45% threshold*: We selected the 45% threshold because it represents the best recovery rate that the other two models achieved when sampling at very low temperatures. To address your suggestion, we have added an additional plot at an intermediate recovery rate cut-off (see Figure 1 in PDF). We have added the unit for speed in Figure 5, which is ‘second’ in the revision. Moreover, Table 3 in PDF compares the sampling speed of our model at varying step sizes against other models. Note that the sampling speed of DDIM is influenced by the choice of skip steps (as illustrated in Figure 5). The inference time of GraDe-IF decreases significantly with increasing step size, and it becomes faster than all baseline methods when step size=100.
> 6. *missing reference*: The claim was from D3PM [1]. We introduced the work in related work and added the reference in line 49  in the revised version.
> 7. *missing definitions*: In the revised version, we: (1) added the full name of FFN in the caption of Figure 1; (2) added the definition of $d$ (line 126), which is the number of amino acid types, i.e., d=20; (3) For the other two notations, $\mathcal{E}$ (the set of edges) and $\mathbf{A}$ (the adjacency matrix), we updated the definition of a graph to $\mathcal{G}=(\mathbf{X}, \mathbf{A}, \mathbf{E})$, with $\mathbf{X}, \mathbf{A}$ and $\mathbf{E}$ representing the node feature matrix, adjacency matrix, and edge feature matrix of the graph, respectively.
> 8. *confusing mathematical notation*: In the revised version, we: (1) unified the notation of graphs by $\mathcal{G}=(\mathbf{X}, \mathbf{A}, \mathbf{E})$ (see the previous answer); (2) updated $\mathbf{X}$ in line 93 to $\mathbf{X}^{pos}$; (3) corrected the definition of $\mathbf{Q}_t$ to $\mathbf{Q}_t = \alpha_t \mathbf{I} + (1 - \alpha_t)\mathbf{1}_d\mathbf{1}_d^{\top}/d$, where $\mathbf{1}_d$ denotes a d-dimensional one vector.
> 9. *figure reference*: Thank you for identifying these typos. We have fixed both references in the revision. In lines 262&289 (of the revised version) the wrong number (“Figure 7”) has been corrected to “Figure 3”.
> 10. *reference to Appendix*: We have added proper references in the revision. For instance, Appendix B (non-Markovian forward process) is now linked in Section 3.4 (DDIM sampling process); Appendix C (algorithm) is mentioned at the beginning of Section 4 (Experiment); and Appendix F (ablation study) is mentioned in Section 4.2.
> 11. *grammar check*: We've conducted a thorough grammar check, correcting various grammatical errors and typos in the revised version.
>
>
> **Reference**:
>
> [1] Austin, Jacob, et al. "Structured denoising diffusion models in discrete state-spaces." Advances in Neural Information Processing Systems 34 (2021): 17981-17993.

---

> > ### Comment · Reviewer_j843 · 2023-08-16
> >
> > I would like to thank the authors for their elaborate response to my feedback and for investing time into generating new results. Here is my answer to the rebuttal:
> >
> > **Questions about new results:**
> >
> > a) Table 4: Can you comment on statistical significance? I think the claim “Our GraDe-IF constantly achieves the *highest TM score* with the smallest variance” might be a bit strong. And why were these 4 proteins selected?
> >
> > b) Great that most of the sequences seem foldable. I was just wondering, is there something the remaining three unfordable sequences have in common? For example, are these perhaps transmembrane, partly disordered, higher beta-sheet content, etc? This would give some insights into potential weaknesses and give some idea as to where you might run into more trouble when exploring the rest of the PDB IDs beyond the 42 you show here.
> >
> > **Weaknesses:**
> > 1. Thank you for spending time and effort to generate new results. Did you come up with this “1-sequence_identity” score or do you have a reference? Moreover, are all generated sequences unique?
> > 2. Much appreciated.
> >
> > **Questions:**
> > 1. See above. In addition, are you planning on adding some more sample visualisations like in Figure 6 for the new proteins, either in the main paper or in the appendix?
> > 2. Okay, fair enough. Maybe worth including in the conclusion / discussion.
> > 3. \- 11. Thank you for making these changes.

---

> > > ### Author Response · Authors · 2023-08-16
> > > **Re-Reviewer j843: response to new questions**
> > >
> > > Thank you for the reply. Regarding your questions:
> > >
> > > **New Results**:
> > >
> > > a). The selection of the four proteins for our new visualizations remains consistent with our initial submission, as they were randomly chosen from the test set. We have now conducted testing on 100 proteins, and the corresponding results are presented in the table below. Notably, GraDe-IF demonstrates a significant performance improvement over PiFold and achieves comparable results to ProteinMPNN. In light of this, we would modify our claim to "achieve the best performance on TM score" as you have suggested.
> > >
> > > | Method     | Success | TM score       |
> > > |-------------|:---------:|:-----------------:|
> > > | PIFOLD     | 85      | 0.80 +- 0.22   |
> > > | ProteinMPNN| 94      | 0.86 +- 0.16   |
> > > | GraDe-IF   | 94      | 0.86 +- 0.17   |
> > >
> > > b). Our investigation into the three 'unfoldable' proteins revealed intriguing insights. These proteins faced challenges not only with our model but also with the baseline models. Notably, the structural determination of the three failed proteins, 1BCT (https://www.rcsb.org/structure/1BCT), 1BHA (https://www.rcsb.org/structure/1bha), and 1CYU (https://www.rcsb.org/structure/1CYU), were all based on NMR, which is an experimental technique that analyzes protein structure in a buffer solution. Due to the presence of multiple structures for a single protein in NMR studies, it is reasonable for folding tools to assign low foldability scores. In addition, we extended our investigation to another protein, 1H3L (https://www.rcsb.org/structure/1H3L), which our model successfully folded but PiFold failed to do so. This particular protein, a fragment of SigR, displays remarkable flexibility, leading to its low experimental resolution that causes a low structure prediction confidence.
> > >
> > >
> > > **Weaknesses**:
> > > 1. We followed [1] when defining the score for diversity. We will include this reference in the final version when discussing the related results.
> > >
> > > **Questions**:
> > > 1. Your suggestion of showcasing the performance of our model on a broader range of proteins is valuable. Given the substantial number of proteins in the test set, including them all might indeed dilute the main results. Instead, we intend to incorporate a selection of representative proteins in the revised appendix. These chosen proteins will exemplify various characteristics, such as those that are unfoldable, exhibit high pLDDT scores, manifest diversity, and more.
> > > 2.  We appreciate your thoughtful suggestion and will certainly include this aspect in the revised version.
> > >
> > > **Reference**:
> > >
> > > [1] Zheng, Zaixiang, et al. "Structure-informed language models are protein designers." bioRxiv (2023): 2023-02.

---

> > > > ### Comment · Reviewer_j843 · 2023-08-17
> > > >
> > > > Thank you for continuously updating your results, I appreciate how much work you are doing in this discussion period.
> > > >
> > > > a) Again, also for these new results, I disagree with “GraDe-IF demonstrates a *significant* performance improvement over PiFold”. The standard deviations for the TM scores are quite large, and all methods are actually “within reach” of one another. It’s fine to not be absolute state of the art, as long as the results are presented in a fair way. It’s still a very interesting method.
> > > >
> > > > b) Very interesting! I would definitely include a discussion about this in the paper / the appendix, to make your story stronger.
> > > >
> > > > All my other concerns were addressed, thank you. I think the exemplary proteins (appendix) will be very interesting to see.

---

### Official Review · Reviewer_YmFn · 2023-07-06

**Soundness:** 3 good
**Presentation:** 3 good
**Contribution:** 3 good
**Rating:** 7
**Confidence:** 3

**Summary:**

This work presents a denoising diffusion model for protein inverse folding: predicting the amino acid sequences that fold into the given 3D protein structure. The proposed method leverages a discrete denoising diffusion model with respect to the graph structure representing the protein backbone. This work proposes to use Blocks Substitution Matrix for the transition matrix considering the different transition probabilities between amino acids, and further utilize the distinct types of secondary structure as a condition during sampling that guides the sampling process to appropriate 3d structures.

**Strengths:**

- The paper is well-written and easy to follow.

- The approach to using (discrete) diffusion models on protein inverse folding is novel to the best of my knowledge.

- Two main contributions are novel and well-motivated: Employing BLOSUM for considering transitions between AAs and using the secondary structure information as a condition during sampling injects biological prior knowledge into the diffusion process which further reduces the sampling space and results in plausible AA sequences.

- The proposed method shows superior performance for the inverse folding tasks in terms of perplexity and recovery rate. The diversity analysis shows that the generated sequences are diverse without losing recovery rate.

**Weaknesses:**

- Using the distinct types of secondary structures as a condition for the sampling process is not clear. How is the information used? Is it an input to the model during each step of the sampling process?

- The reason for not giving a higher rating is that although the approach to the task is novel, using BLOSUM and secondary structure conditions are novel, and the method shows superior performance, the components of the proposed methods are widely used in different domains: Diffusion models and E3 equivariant networks are widely used in the generation of protein structures given sequences (I understand that the task is not the same but in the similar domain) and DDIM is also used to reduce the sampling steps.

- Minor: line 143 "orm.alized"

**Questions:**

- Using DDIM for sampling, how long does it take to predict the sequence compared to the baselines, e.g. ProteinMPNN or ESM-IF1?

**Limitations:**

The limitations are discussed in the Supplementary file.

---

> ### Author Rebuttal · Authors · 2023-08-08
>
> We sincerely appreciate your thoughtful feedback on our work, particularly your kind recognition of the quality of our presentation, the originality of our work, and its significance. We would like to address your questions and concerns as follows:
>
> **Weaknesses**:
> - *secondary structure representation*: In our approach, we employ an eight-dimensional one-hot encoded feature to represent nodes (amino acids) associated with one of the eight secondary structures. During the reverse diffusion process, this secondary structure information undergoes projection into the feature dimension and is subsequently integrated into the time embedding. This embedding is then added to the features of the output of each layer.
> - *novelty of the work*: We agree that many individual components, such as the forward/reverse diffusion model, equivariant graph neural networks, and DDIM, have been previously explored in research. However, what sets our work apart is the non-trivial combination of these components, achieved through significant additional effort. Notably, we derive a substitution matrix for multi-step transitions in discrete space, leading to a closed-form expression for this matrix, which in turn allows us to establish the Discrete Discrete-time Inhomogeneous Markov (DDIM) model for protein sequence generation. Furthermore, our approach's effectiveness in generating reliable protein sequences is attributed to the careful arrangement of auxiliary components, including the transition matrix for forward/reverse diffusion and the incorporation of protein secondary structures for conditional sampling. We contend that the overall framework we introduce to address the inverse folding problem, encompassing evaluation metrics and related aspects, constitutes an additional valuable contribution.
> - minor typo: Thank you for pointing out this typo. We have corrected it to “normalized” in the paper.
>
> **Questions**:
> - *sequence prediction time*: The sampling speed of DDIM is influenced by the choice of skip steps (as illustrated in Figure 5). To illustrate, we present the generation of 3FKF-A and provide a comparative analysis of inference times along with baseline methods, considering skip steps of 100, 20, and 1 for GraDe-IF, where the skip step of 1 reverts to the original DDPM sampling algorithm. The inference time of GraDe-IF decreases significantly with increasing step size, and it becomes faster than all baseline methods when step size=100.
> | Method                       | Inference Time (in seconds)  |
> |------------------------------|:--------------------------------------------:|
> | PiFold         	        | 0.06                                              |
> | ProteinMPNN    	        | 0.14                                              |
> | ESM-if1           	        | 0.17                                              |
> | GraDe-IF (step=1)     | 6.14                                              |
> | GraDe-IF (step=20)   | 0.31                                              |
> | GraDe-IF (step=100) | **0.05**                                              |

---

> > ### Comment · Reviewer_YmFn · 2023-08-17
> >
> > Thank you for the detailed response.
> > I agree with the authors that the novel combinations of some known approaches as in this case provide a simple yet effective method. I do not find other main concerns and would like to keep my score.
> > I tend to accept this work if there is no main issue commented on by other reviewers in the experimental part.

---

### Author Rebuttal · Authors · 2023-08-09

We thank all the reviewers for their detailed comments and insightful suggestions. We incorporated additional experiments and analyses as per recommendations. Here, we present a concise overview of the major enhancements that have been universally implemented, focusing on aspects such as diversity (Figure 1, Table 1), structural comparison (Table 2), sampling speed (Table 3), and foldability (Table 4 & 5). The visual representation in the form of the figure and tables has been included within the attached PDF, with corresponding references provided in each of our responses. We believe that a brief overview of these additional results will provide a clear context for comprehending the significance of the updates we have made.

 **Figure 1. t-SNE of the generated sequences in three different recovery levels.**

In response to the task of aligning recovery rate levels, we have introduced an intermediary plot positioned between the two subfigures in Figure 4. This supplementary plot effectively illustrates the intricate relationship between recovery rates and diversity across the three methods on a finer scale. Notably, the visualization highlights the rapid contraction of the sampling space for both PiFold and ProteinMPNN. In contrast, the samples generated by GraDe-IF exhibit a broader distribution that encompasses the wild type.

**Table 1. Comparison of diversity and recovery rate at three different levels**

The recovery rate and diversity of the generation results were compared by controlling temperature = {0.5, 0.1, 0.0001} for PiFold and ProteinMPNN and sample step={20, 100, 250} for GraDe-IF. The three levels are divided roughly into low, medium, and high recovery rates in the table below. The results, quantified as average diversity and recovery rate, are depicted in the table provided. For clarity, the metric "diversity" is quantitatively defined as `diversity = 1-sequence identity`, where the average sequence identity is computed pairwise for generated sequences. For instance, in the first cell of the table, where the method is PiFold, the recovery level is low, and the metric is diversity, the diversity value is indicated as 0.3796.

**Table 2. Average RMSD and pLDDT across 42 protein structures (folded by AlphaFold2) with model-generated sequences.**

We expanded the validation of RMSD and pLDDT across a wider range of 42 proteins. For each protein backbone, a sequence is generated per model, which is then folded by AlphaFold2 to calculate the average pLDDT and RMSD with respect to the template wild-type protein structure. Notably, GraDe-IF exhibited the highest average pLDDT and lowest RMSD, highlighting its effectiveness over baseline methods.

**Table 3. Sampling speed of GraDe-IF (DDIM) at varying step sizes and baseline methods.**

The sampling speed of DDIM is influenced by the choice of skip steps (as illustrated in Figure 5). To illustrate, we present the generation of 3FKF-A and provide a comparative analysis of inference times along with baseline methods, considering skip steps of 100, 20, and 1 for GraDe-IF, where the skip step of 1 reverts to the original DDPM sampling algorithm. The inference time of GraDe-IF decreases significantly with increasing step size, and it becomes faster than all baseline methods when step size=100.

**Table 4. TM score comparison. 10 sequences were generated for each protein backbone.**

We computed the TM score to measure the foldability associated with the sequences generated by gauging the structural similarity between the structure of the generated sequences and the native structures. Generally speaking, a higher TM score demonstrates a better chance for the structure to be folded, and a protein with a TM score lower than 0.5 is basically considered unfoldable. The table below summarizes the average TM scores (along with the standard deviations) between the structures predicted by AlphaFold2 for generated sequences and their associated wild-type structures. Our GraDe-IF constantly achieves the highest TM score with the smallest variance.

**Table 5. TM score comparison on 42 protein backbones, with each backbone generating 1 sample. A sequence is considered foldable if its TM score > 0.5.**

The foldability of the generated protein sequences was also evaluated on a wider range of 42 proteins (Note that the reason we not investigating a larger volume of proteins is because that AlphaFold failed to fold more structures within the limited time for rebuttal). We define the number of successfully folded proteins by whether their TM score is larger than 0.5. Again, GraDe-IF delivers the largest amount of foldable proteins with the highest average TM score.

---

> ### Comment · Reviewer_92pD · 2023-08-10
> **Clarity on choice of 42 proteins**
>
> Can the authors clarify how these 42 proteins were chosen? How do we know these proteins were not cherry picked?

---

> > ### Author Response · Authors · 2023-08-13
> > **Re: Clarification on the reported 42 proteins**
> >
> > We fold the first 42 proteins in the test dataset in alphabetical order by their PDB ID. Due to time limitations, we were only able to fold such a small subset of proteins' structures for the generated protein sequences by the three models. We are currently working on conducting evaluations on the remaining proteins in the test set. We are committed to providing updated and comprehensive scores for a broader range of proteins.

---

### Decision · Program_Chairs · 2023-09-21

**Decision:**

Accept (poster)

**Comment:**

There was agreement among the reviewers that the authors method is novel, solves an important problem, and achieves very competitive performance. The reviewers mostly suggested a variety of new experiments, and the authors provided these results extensively, which they should incorporate into the camera ready version of the paper. Overall, the authors' performance improvements over even very recent inverse folding methods like PiFold are impressive, and the approach taken seems novel.